# Optimization of the Vacuum Microwave Assisted Extraction of the Natural Polyphenols and Flavonoids from the Raw Solid Waste of the Pomegranate Juice Producing Industry at Industrial Scale

**DOI:** 10.3390/molecules26041033

**Published:** 2021-02-16

**Authors:** Konstantinos Petrotos, Ioannis Giavasis, Konstantinos Gerasopoulos, Chrysanthi Mitsagga, Chryssoula Papaioannou, Paschalis Gkoutsidis

**Affiliations:** 1Department of Agrotechnology, School of Agricultural Sciences, Geopolis Campus, University of Thessaly, Periferiaki Odos Larisas Trikalon, 41500 Larisa, Greece; kosgera1@yahoo.gr (K.G.); xrpapa@teilar.gr (C.P.); gkoutsidis@teilar.gr (P.G.); 2Department of Food Science and Human Nutrition, School of Agricultural Sciences, Karditsa Campus, University of Thessaly, Terma Odou N. Temponera, 43100 Karditsa, Greece; igiavasis@teilar.gr (I.G.); cmitsanga@uth.gr (C.M.)

**Keywords:** pomegranate pomace, vacuum microwave assisted extraction (VMAE), modified response surface optimization, historical data design, polyphenols, flavonoids, industrial scale optimization

## Abstract

Pomegranate pomace (PP) is the solid waste produced in bulk by the pomegranate juice industry which is rich in polyphenols and flavonoids that can replace the hazardous chemical antioxidants/antimicrobials currently used in the agro-food and cosmetics sectors. In the present work, the vacuum microwave assisted extraction (VMAE) of natural antioxidants from raw pomegranate pomace was investigated and successfully optimized at an industrial scale. For the optimization of PP VMAE a novel, highly accurate response surface methodology (RSM) based on a comprehensive multi-point historical design was employed. The optimization showed that the maximum recovery of PP total polyphenols as well as total PP flavonoids were obtained at microwave power = 4961.07 W, water to pomace ratio = 29.9, extraction time = 119.53 min and microwave power = 4147.76 W, water to pomace ratio = 19.32, extraction time = 63.32 min respectively. Moreover, the optimal VMAE conditions on economic grounds were determined to be: microwave power = 2048.62 W, water to pomace ratio = 23.11, extraction time = 15.04 min and microwave power = 4008.62 W, water to pomace ratio = 18.08, extraction time = 15.29 min for PP total polyphenols and PP total flavonoids respectively. The main conclusion of this study is that the VMAE extraction can be successfully used at industrial scale to produce, in economic manner, high added value natural extracts from PP pomace.

## 1. Introduction

Pomegranate fruit (*Punica granatum* L.) production is a fast-growing agricultural activity as the fruit is globally recognized as a “superfood”, due to its nutritious characteristics. The global pomegranate market was valued at 8.2 billion USD in 2018 and is expected to reach 23.14 billion USD by year 2026 [1], while the global pomegranate fruit production runs into 3,000,000 MT [2]. In a recently published paper [3] on the optimization of the microwave-assisted extraction of pomegranate pomace (PP) at a lab scale, the topic of the sustainable utilization of PP was thoroughly investigated and it is cited hereafter. According to Damian [4], the pomegranate, which originated in the Middle East and India and has been used for centuries in ancient cultures for its medicinal properties, is an important fruit of tropical and subtropical regions. It is also widely reported that pomegranate exhibits antiviral, antioxidant, anticancer, and anti-proliferative activities [5,6,7,8].

Pomegranate fruit contains valuable antioxidants and according to Li et al. [9] the polyphenolic content of pomegranate juice is higher when this is produced from the whole fruit instead of only the arils. This indicates that there is a considerable phenolic content in the pomegranate peel, as well as in the solid pomegranate pomace, which is a by-product of the pomegranate juice industry and represents about 50% of the total processed fruits. As cited by Li et al. [9], Fischer et al. [10] and Saad et al. [11], the profile of polyphenolic content of the pomegranate pomace contains polyphenols, flavonoids, proanthocyanidins, hydrolysable tannins (like ellagic acid, pedunculagin, punicalin and gallic acids) in substantial amounts, ranging from 27 g kg^−1^ to 172 g kg^−1^ of dry pomace, expressed as gallic acid equivalents. Furthermore, in a research paper published by Elfalleh et al. [12] the total polyphenols content of pomegranate pomace (expressed as gallic acid) was found to be 85.60 ± 4.87 mg g^−1^. According to Farag et al. [13] and Dimou [14] the primary polyphenols contained in pomegranate pomace are gallic acid, proto-catechuic acid, chlorogenic acid, vanillic acid, coumarin, caffeic acid, oleuropein, ferulic acid, quercetin and caffeine.

Nowadays, natural antioxidants and antimicrobials have become very popular for novel food/nutraceuticals, cosmetics and phytoprotection applications and are preferred by consumers over synthetic antioxidants, such as butylhydroxyanisole (BHA) and dibutylhydroxytoluene (BHT) or propyl gallate (PG) [15,16,17,18] or synthetic preservatives like sorbate salts and chemical pesticides in agricultural applications. Besides, avoiding the undesirable health effect of some synthetic chemicals, the use of natural alternatives of antimicrobials and antioxidants from pomegranate can have beneficial health effects. For example, enrichment of ice cream with pomegranate by-products resulted in increased phenolic content of ice creams, which caused an improvement in antioxidant and anti-diabetic activities, mainly due to the functional properties of punicalagins in pomegranate peel, and punicic acid in pomegranate seed oil [19]. Furthermore, addition of pomegranate to popular chicken meat products enhanced its shelf life by 2–3 weeks during chilled storage [20]. In addition, the enhanced antioxidant activity of pomegranate peel extract was found to inhibit lipid oxidation in cooked chicken patties [21].

A recent literature review [22] cited several studies that have reported on in vitro bioactivity of pomegranate peel extracts, including antioxidant, antitumor, anti-inflammatory, and anti-proliferative properties. Kanatt et al. [20] investigated the antioxidant and antimicrobial potential of pomegranate peel extract (PPE) and concluded that the efficacy of PPE in scavenging hydroxyl and superoxide anion radicals was very high. In addition, the extract had good reducing power and iron chelation capacity and showed good antimicrobial activity against *Staphylococcus aureus* and *Bacillus cereus*, having a minimum inhibitory concentration of only 0.01%. *Pseudomonas species* could be also inhibited at a higher concentration of 0.1%, while PPE was ineffective against *Escherichia coli* and *Salmonella typhimurium.* Thus, PPE could potentially be included in several industrial products (e.g., as ingredient in functional foods), due to its versatile functional properties. After addition of PPE at a concentration of 800–850 ppm in sunflower oil [23] and 200–1000 ppm in fish oil [24] high stabilization efficiency was exhibited, which was comparable to that achieved by conventional synthetic antioxidants (i.e., BHT used at its maximum allowed concentration). Similarly, Kumudavally et al. [25] and Devatkal et al. [26] reported that PPE significantly increases the stability of beef and goat meat products against lipid peroxidation. Furthermore, addition of PPE to jams [27], juices and wines [28] increased their phenolic, flavonoid, and thiol concentration with a significant improvement of the free radical scavenging and product stability features. In addition, Kaderides et al. [29], incorporated pomegranate peel extract in hazelnut paste and reported an inhibition of lipid oxidation with reduced formation of peroxides.

Many more references in the literature point out the potential of pomegranate pomace or peel extract to replace synthetic antioxidants and antimicrobials. Its exceptional bioactivity is largely attributed to the presence of punicalagin, one of the main polyphenols of pomegranate peel [30,31,32,33].

The production of high quality bioactive natural extracts depends on the extraction method and on the conditions that maximize the concentration of the bioactive compounds in the final extract. For this reason, new eco-green extraction methods have been used and optimized in order to produce effective, natural extracts from organic agro-food byproducts like pomegranate pomace [4,22,34]. The main “green” extraction technologies which are nowadays available in the market at reasonable price are microwave-assisted extraction and ultrasound-assisted extraction, which have certain advantages, compared with conventional extraction methods, such as: less consumption of solvent, better retention of the bioactivity of the extracted polyphenols, lower operation temperatures and less energy consumption [35,36]. These two technologies can also involve operation under vacuum, which is preferable for preserving the bioactivity of the polyphenols and prevents their oxidative degradation during the extraction process, thus yielding an extract of high quality. Kaderides et al. [22] suggested that, between these two “green” extraction technologies microwave technology is more advantageous compared to ultrasound technology, since it can provide 1.7 times higher polyphenol concentration in the extract in about half the time needed for ultrasound assisted extraction. Just recently, Skenderidis et al. [37] investigated and optimized the VMAE of raw pomegranate peel (part of pomegranate pomace which include only the outer layer of the fruit). In this research work by Skenderidis et al. [37] conventional RSM methodology based on classical Box and Benhken experimental design was employed.

The target of the present work is to investigate and Optimize the vacuum microwave extraction of raw pomegranate pomace, for the first time at industrial scale, and with dual maximum recovery as well as maximum economic performance criteria and thus obtain the optimum extraction conditions (microwave power, water to raw PP ratio, extraction time) corresponding to each one, respectively, of the above mentioned targets.

## 2. Results

### 2.1. Predictive Modeling and Optimization of the Extracted Amount of PP Total Polyphenols

The results of the total polyphenol content of the pomegranate extracts are presented in Table 1. In particular, 73 vacuum microwave-assisted extraction experiments were carried out and three samples of the extract were collected in each respective run. The total polyphenol contents of each one of the obtained three samples per run were determined and the average values of them are listed in Table 1. By using the data of the extracted PP total polyphenols and thereafter by applying modeling to them by response surface methodology (RSM; selected options: historical data experimental design, stepwise regression and cubic polynomial RSM model) the R^2^ value for the model was found to be 0.8997. In order to improve the correlation between TPE (mg GAE) and the three extraction factors (microwave power (W), water/raw PP ratio and extraction time (min)) the TPE values were divided by t (extraction time in min) and the obtained values of the new, modified response TPE/t (mg GAE min^−1^) are also listed in Table 1. Consequently, the data for TPE/t were statistically analyzed by RSM using the options: historical data experimental design, stepwise regression and adaptation to cubic polynomial RSM model to derive the model equation. In addition, the relevant model statistics were obtained by analysis of variance (ANOVA).

The ANOVA results, which are summarized in Table 2, proved that the derived cubic polynomial model which correlates the new modified TPE/t response to the extraction factors: (a) microwave power (W), (b) water to raw pomace ratio and (c) extraction time (min) was significant while its lack of fit was found to be insignificant. Furthermore, as it was suggested by Cox & Box plot, the natural log function had to be selected as the most appropriate response modification with target to achieve high model accuracy.

From the fit statistics of the ANOVA it is concluded that the R^2^ value is equal to 0.9858 which means that the modified response TPE/t is very well correlated with the experimental factors A = microwave power (W), B = water to raw PP pomace ratio and C = extraction time in (min). On the other hand, the obtained values of adjusted R^2^ = 0.9824 and predicted R^2^ = 0.9740 were found to be very close to each other and their difference was, by far, less than the maximum allowed value of 0.2 which is demanded in order to have a reliable model capable to be used for accurate predictions within the limits of the selected design space [38].

The final equation for modified TPE/t response (Εquation (1)) is:(1)$$\begin{array}{cc} \mathrm{Ln}\left(\frac{\mathrm{TPE}}{t}\right)= & +7.18681\;+\;\left(3.70306\times 10^{-4}\right)\times A+0.21905\times B-0.058200\times C-\left(3.34433\times 10^{-5}\right)\times A\times B\\ & +\left(1.29273\times 10^{-6}\right)\times A\times C-\left(2.84000\times 10^{-4}\right)\times B\times C+\left(4.12548\times 10^{-9}\right)\times A^{2\;}\\ & -\left(4.94457\times 10^{-3}\right)\times \;B^{2\;}+\;\left(4.98042\times 10^{-4}\right)\times \;C^{2\;}+\;\left(9.33915\times 10^{-8}\right)\times A\times \;B\times C\\ & -\left(1.70747\times 10^{-9}\right)\times A^{2\;}\times B+\left(9.59174\times 10^{-7}\right)\times A\times B^{2\;}-\left(1.76838\times 10^{-8}\right)\times A\times C^{2\;}\\ & -\left(1.49358\times 10^{-6}\right)\times C^{3\;} \end{array}$$

Moreover, in Appendix A the predicted vs. actual values of the TPE/t response are illustrated and from the proximity of the graph points to the central 45° line a very good fit of the derived model to the experimental data was concluded. In addition, by examining the model by a series of criteria given by Design Expert 7.0.0 Software (Stat-Ease Inc., Minneapolis, USA), it was also pointed out that this was well in the limits set by them. A typical example is given in Appendix A, where the Externally Studentized residuals plot is presented and all the points are within the limits providing thus an additional proof for the effectiveness of the model predictions as no outliers (points out of the limits set in the plot) are existent.

By re-arranging Equation (1) presented in Table 2 (multiplying both sides by the extraction time t) we get the model equation for TPE (mg GAE) which has the following form:TPE (mg GAE) = C × e^F(A,B,C)^(2)
where: A = microwave power (W), B = water to raw PP ratio, C = extraction time (min) and:(3)$$\begin{array}{cc} F\left(A,B,C\right)\;= & +7.18681\;+\;\left(3.70306\times 10^{-4}\right)\times A\;+\;0.21905\times B-0.058200\times C\\ & -\left(3.34433\times 10^{-5}\right)\times A\times B\;+\;\left(1.29273\times 10^{-6}\right)\times A\times C-\left(2.84000\times 10^{-4}\right)\times B\times C\;\\ & +\;\left(4.12548\times 10^{-9}\right)\times A^{2\;}-\left(4.94457\times 10^{-3}\right)\times \;B^{2\;}+\;\left(4.98042\times 10^{-4}\right)\times \;C^{2\;}\\ & +\;\left(9.33915\times 10^{-8}\right)\times A\times \;B\times C-\left(1.70747\times 10^{-9}\right)\times A^{2\;}\times B\\ & +\left(9.59174\times 10^{-7}\right)\times A\times B^{2\;}-\left(1.76838\times 10^{-8}\right)\times A\times C^{2\;}-\left(1.49358\times 10^{-6}\right)\times C^{3} \end{array}$$

By substituting in Equation (2) the A, B, C values from the experimental plan and listed in Table 1, the response TPE was recalculated and the new data set was introduced in the Design Expert software and RSM modeling was applied. The derived RSM model was of cubic polynomial form and the appropriate transformation of the response (TPE) according to the Cox & Box plot was the natural log function. In addition, following the relevant fit statistics the R^2^ value was found to be equal to 0.9877 which is considered very high while the values of adjusted R^2^ = 0.9847 and predicted R^2^ = 0.9810 were very close to each other and its difference was, by far, lower than the proposed maximum of 0.2 which is considered to be the higher accepted limit for satisfactory simulation [38]. Finally, the soundness of the model described by Equation (2) was checked and confirmed by using additional statistical RSM criteria like: (i) actual vs. predicted values of TPE; (ii) externally Studentized residuals; (ii) DFFTS vs. run; (iv) residuals vs. run. The study of the plots which are summarized in Appendix A and correspond to the reported statistical criteria dictates that (i) in the predicted vs. actual value plot the correlation points are lying very close to the central line (45°) of the plot, (ii) the points in externally studentized residuals plot are well within the limits which means that there are not outliers (points far from the model prediction) and finally (iii) the correlation points in DFFTS vs. run and (iv) the residuals vs. run plots are also well within the limits set by statistics. In addition, according to the conducted analysis of variance (ANOVA) the derived RSM model is significant and its lack of fit is not significant and the significant model terms and interactions are the following: A, B, AB, AC, BC, A^2^, B^2^, C^2^, ABC, A^2^B, AB^2^, AC^2^, C^3^.

Moreover, Figure 1, illustrates the interactions between the factors A, B, C and their effect on TPE response.

Finally, by using the optimization routine of the Design Expert software the optimized value of the TPE (mg GAE 2 kg^−1^) as well as the corresponding optimal values of the extraction factors were obtained and are presented hereafter:A = Microwave power (W) = 4961.07 WB = Water to raw PP ratio = 29.90C = Extraction Time (min) = 119.53Maximum TPE (mg GAE 2 kg^−1^ raw PP) = 138,404 mg GAE 2 kg^−1^ raw PP or equivalently: 69,202 mg GAE kg^−1^ of raw PP.

However, taking into account that the moisture of the raw pomegranate pomace used in this work was determined by the classical ASTM method and found to be 67% the maximum amount of TPE value expressed on dry basis is calculated as: 69,202 mg GAE kg^−1^ of raw PP/0.33 = 209,703 mg GAE kg^−1^ of dry PP.

### 2.2. Predictive Modeling and Optimization of the Extracted Amount of PP Total Flavonoids (TFE)

The results of the total flavonoids content of the pomegranate extracts (mg QE) are presented in Table 1. In particular, 73 vacuum microwave extraction experiments were carried out and three samples of the extract were collected in each respective run. The total flavonoids contents of each one of the obtained three samples per run were determined and the average values of them are listed in Table 1. By using the data of the extracted PP total flavonoids and thereafter by applying modeling to them by Surface Response (RSM) methodology (selected options: historical data experimental design, stepwise regression and cubic polynomial RSM model) the R^2^ value for the model was found to be 0.7615. In order to improve the correlation between TFE (mg QE) and the three extraction factors (microwave power (W), water/raw PP ratio and extraction time (min)) the TFE values were divided by t^2^ (extraction time squared in min^2^) and the obtained values of the new, modified response TFE/t^2^ (mg QE min^−2^) are listed in Table 1. Consequently, the data for TFE/t^2^ were analyzed by RSM using the options: historical data experimental design, stepwise regression and cubic polynomial RSM model in order to derive the model equation and obtain the relevant model statistics by analysis of variance (ANOVA).

The ANOVA results, which are summarized in Table 3, proved that the derived cubic polynomial model which correlates the TPE/t^2^ with the extraction factors: (a) microwave power (W), (b) water to raw pomace ratio and (c) extraction time (min) was significant while its lack of fit was insignificant. Furthermore, as it is suggested by Cox & Box plot, the natural log function was the most appropriate modification of the TFE/t^2^ response with target to obtain the optimum model accuracy.

From the ANOVA fit statistics it was concluded that the R^2^ value is equal to 0.9693 which means that the modified response TFE/t^2^ is highly correlated with the experimental factors A = microwave power (W), B = water to raw PP pomace ratio and C = extraction time in (min). On the other hand, the values of adjusted R^2^ = 0.9631 and predicted R^2^ = 0.9510 found to be very close to each other and their difference is, by far, less than the maximum allowed value of 0.2 which is demanded in order to have a reliable model capable to be used for accurate prediction within the limits of the design space [38]. Moreover, by studying the model evaluation plots provided by Design Expert ANOVA statistics and in particular predicted vs. actual values plot, externally Studentised residuals plot, residual vs. run plot and DFFITS plot, it is concluded that all the evaluation criteria were successfully met. This provides an additional proof for the effectiveness of the model predictions, as no outliers (bad prediction points, out of the limits set in the plot) are existent.

Final Equation for modified TFE/t^2^ response:(4)$$\begin{array}{cc} \mathrm{Ln}\frac{\mathrm{TFE}}{t^{2}}\;= & -0.33411\;\\ & +\;(1.54632\times 10^{-3})\times A+0.35967\times B-0.10865\times C-\left(5.13118\times 10^{-5}\right)A\times B\\ & -\left(2.14116\times 10^{-6}\right)\times A\times C-\left(2.58780\times 10^{-4}\right)\times B\times C-\left(1.66226\times 10^{-7}\right)\times A^{2}\\ & -\left(6.58994\times 10^{-3}\right)\times \;B^{2}+(1.00011\times 10^{-3})\;\times C^{2}+\;\left(1.41188\times 10^{-7}\right)\times A\times B\times C\;\\ & +\left(4.91944\times 10^{-9}\right)\times A^{2}\times B\;-\left(3.79856\times 10^{-6}\right)\times C^{3} \end{array}$$

By re-arranging the model Equation (4) presented in Table 3 (multiplying both sides by the extraction time squared, t^2^) we get the model equation for TFE (mg QE) which has the following form:TFE (mg QE) = C^2^ × e ^f(A,B,C)^(5)
where: A = microwave power (W), B = water to raw PP pomace ratio and C = extraction time (min) and:(6)$$\begin{array}{cc} f\;\left(A,\;B,\;C\right)\;= & -0.33411\;\\ & +\;(1.54632\times 10^{-3})\times A+0.35967\times B-0.10865\times C-\left(5.13118\times 10^{-5}\right)\times A\times B\\ & -\left(2.14116\times 10^{-6}\right)\times A\times C-\left(2.58780\times 10^{-4}\right)\times B\times C-\left(1.66226\times 10^{-7}\right)\times A^{2}\\ & -\left(6.58994\times 10^{-3}\right)\times \;B^{2}+\;\left(1.00011\times 10^{-3}\right)\times \;C^{2}+\;\left(1.41188\times 10^{-7}\right)\times A\times B\times C\;\\ & +\left(4.91944\times 10^{-9}\right)\times A^{2}\times B\;-\left(3.79856\times 10^{-6}\right)\times C^{3} \end{array}$$

By substituting in Equation (5), the A, B, C values included in the experimental plan and listed in Table 1, the response TFE was recalculated and the new data set was introduced in the Design Expert software and RSM modeling was applied. The derived RSM model was of cubic polynomial form and the appropriate transformation of the response (TFE) according to the Cox & Box plot was the natural log function. In addition, according to the fit statistics the R^2^ value was found to be equal to 0.9901 which is satisfactory while the values of adjusted R^2^ = 0.9881 and predicted R^2^ = 0.9855 found to be very close to each other and its difference adequately lower than the proposed maximum 0.2 which is the higher accepted limit to claim satisfactory simulation [38]. Finally, the soundness of the model described by Equation (5) was checked and confirmed by using the additional statistical criteria proposed by the RSM theory and in particular: (i) actual vs. predicted values of TFE (ii) externally Studentized residuals (iii) DFFTS vs. run (iv) residuals vs. run. From the study of the relevant plots, which are summarized in Appendix A and correspond to the previously reported statistical criteria it was shown that (i) in the predicted vs. actual value plot, the correlation points are lying very close to the central line (45°) of the plot, (ii) the points in externally studentized residuals plot are well within the limits which means that there are not outliers (points far from the model prediction) and finally (iii) the correlation points in DFFTS vs. run and residuals vs. run plots are also well within the limits set by RSM statistics. In addition, according to the conducted analysis of variance (ANOVA) the derived RSM model is significant and its lack of fit is not significant and furthermore the significant model terms and interactions are the following: A, B, AB, AC, BC, A^2^, B^2^, C^2^, ABC, A^2^B, AB^2^, AC^2^, C^3^.

Figure 2 illustrates the interactions between the factors A, B, C and their effect on TFE response.

Finally, by using the RSM optimization routine the optimized value of the amount of TFE (mg QE 2 kg^−1^) as well as the corresponding optimal values of the extraction factors were obtained and are presented hereafter:A = Microwave power (W) = 4147.76 WB = Water to raw PP ratio= 19.32C = t= Extraction Time (min) = 63.32 minMaximum TFE (mg QE 2 kg^−1^ raw PP pomace) = 14,479.3 mg QE 2 kg^−1^ raw PP pomace or equivalently: 7239.65 mg QE kg^−1^ of raw PP pomace.

However taking into account that the moisture of the raw pomegranate pomace used in this work was determined by the classical ASTM method and found to be 67% the maximum amount of TPE value expressed on dry basis is calculated as: 7239.65 mg QE kg^−1^ of raw PP/0.33 = 21,938.3 mg QE kg^−1^ of dry PP.

Furthermore, taking advantage of the facility of the Design Expert optimization capability which allows simultaneous optimization of PP total polyphenols and PP total flavonoids, the optimal PP VMAE extraction parameters were determined as well as the corresponding to them optimum values of the two simultaneously optimized antioxidant parameters:A = Microwave power (W) = 3807.85 WB = Water to raw PP ratio = 21.68C = Extraction Time (min) = 64.5 min.

The optimal values of PP total polyphenols obtained at the abovementioned conditions are:Optimum of PP total polyphenols = 126,224 mg GAE 2 kg^−1^ raw PP = 63,112 mg GAE kg^−1^ raw PPOptimum of PP total flavonoids = 13,799.2 mg GAE 2 kg−1 raw PP = 6899.6 mg QE kg^−1^ raw PP
or on a dry basis: 191,248.5 mg GAE kg^−1^ dried PP and 20,907.88 mg QE kg^−1^ dried PP respectively or expressed as % of the individual optima of PP Total polyphenols and PP total flavonoids 91.2% and 95.3%.

### 2.3. Economic Optimization of the PP VMA Extraction at an Industrial Scale and Determination of the Corresponding Optimum Extraction Condition Values to Obtain the Maximum Rate of Extraction (Productiviy) for PP Total Polyphenols and PP Total Flavonoids, Respectively

#### 2.3.1. Maximization of the Rate of the Extraction (Productivity) of Raw PP Total Polyphenols

In order to maximize the rate of VMAE extraction of PP polyphenols, the modified values of the PP total polyphenols along with the Equation (11) were used. Thus, the rate of extraction of PP was calculated for all points of the experimental domain and correlated to the experimental factors. Consequently, the RSM methodology was followed in order to develop the predictive model and proceed to the optimization (maximization) of the PP total polyphenols extraction rate.

The derived RSM polynomial model was of cubic form and by following the suggestion of Cox & Box plot the appropriate transformation of the response (rate of extraction of PP total polyphenols) was the natural log function. Furthermore, according to the RSM fit statistics the R^2^ value found to be 0.9990 whereas adjusted R^2^ = 0.9987 and predicted R^2^ = 0.9984. This means a very good fitting of the data by the model as R^2^ was almost unit and the difference between adjusted R^2^ and predicted R^2^ was, by far, lower than the statistically set higher limit of 0.2 [38].

The derived RSM model is presented by the following equation:(7)$$\begin{array}{cc} \mathrm{Ln}\left(\mathrm{RATE}\;\mathrm{TPE}\right)= & +6.08301+\left(3.79237\times 10^{-4}\right)\times A+0.22386\times B-0.027881\times C-\left(3.45880\times 10^{-5}\right)\times A\times B\\ & +\left(1.17605\times 10^{-6}\right)\times A\times C-(2.85996\times 10^{-4})\;\times B\times C+\;(4.44141\times 10^{-9})\;\times A^{2}\\ & -\left(5.05688\times 10^{-3}\right)\times B^{2}+\left(1.56775\times 10^{-4}\right)C^{2}+\left(9.35533\times 10^{-8}\right)\times A\times B\times C-\left(1.69328\times 10^{-9}\right)\times A^{2}\times B\\ & +\left(9.84957\times 10^{-7}\right)\times A\times B^{2}-\left(1.69029\times 10^{-8}\right)\times A\times C^{2}-\left(2.01934\times 10^{-7}\right)\times C^{3} \end{array}$$
and the effect of the interactions of the A, B, C extraction factors on the RATE TPE response are presented in Figure 3.

In addition, the optimum extraction conditions and the corresponding to them maximum value of the rate of PP total polyphenols VMAE were found to be equal to:A = Microwave power (W) = 2048.62 WB = Water to raw PP ratio = 23.11C = Extraction Time (min) = 15.04 minValue of maximum rate of PP total polyphenols VMAE = 3782.67 mg GAE kg^−1^ min^−1^ raw PP

The above mentioned extraction parameters reflect to the most economic operation at industrial scale, concerning PP total polyphenols extraction.

#### 2.3.2. Maximization of the Rate of the Extraction (Productivity) of Raw PP Total Flavonoids

In order to maximize the rate of VMAE extraction of PP total flavonoids, the modified values of the PP total flavonoids and Equation (12) were used. Thus, the rate of extraction of PP total flavonoids was calculated for all points of the experimental domain and correlated to the experimental factors. Consequently, the RSM methodology was applied in order to develop the predictive model and then proceed to the optimization (maximization) of the PP total flavonoids extraction rate.

The derived RSM polynomial model was of cubic form and by following the suggestion of Cox & Box plot the appropriate transformation of the response (rate of extraction of PP total flavonoids) was the natural log function. Furthermore, according to the RSM fit statistics the R^2^ value found to be 0.9966 whereas adjusted R^2^ = 0.9960 and predicted R^2^= 0.9951. This means a very good fitting of the data by the model as R^2^ was almost unit and the difference between adjusted R^2^ and predicted R^2^ was, by far, lower than the statistically set higher limit of 0.2 [38].

The derived RSM model is presented by the following equation:(8)$$\begin{array}{cc} \mathrm{Ln}\left(\mathrm{RATE}\;\mathrm{OF}\;\mathrm{TFE}\right) & \\ & =+0.46350+\left(\;1.53978\times 10^{-3}\right)\times A+\;0.36063\times B-0.011730\times C\;\\ & -\left(5.13776\times 10^{-5}\right)\times A\times B-\left(2.11289\times 10^{-6}\right)\times A\times C-\left(2.49809\times 10^{-4}\right)\times B\times C\\ & -\left(1.65825\times 10^{-7}\right)\times A^{2}-\left(6.61766\times 10^{-3}\right)\times B^{2}+\left(4.00917\times 10^{-5}\right)\times C^{2}\\ & +\left(1.38156\times 10^{-7}\right)\times A\times B\times C\;+4.95763\times 10^{-9}\;\times A^{2}\times B-\left(2.83569\times 10^{-7}\right)\times C^{3} \end{array}$$
and the effect of the interactions of A, B, C of the extraction factors on the RATE TFE response are presented in Figure 4.

In addition, the optimum extraction conditions and the maximum value of the rate of PP total flavonoids VMAE corresponding to them were determined to be the following:A = Microwave power (W) = 4008.62 WB = Water to raw PP ratio = 18.08C = t = Extraction Time (min) = 15.29 minValue of maximum Rate of PP total flavonoids VMAE = 339.869 mg QE kg^−1^ raw pomace min^−1^

The above mentioned optimal extraction parameters reflect to the most economic operation at industrial scale, regarding production of PP flavonoids.

### 2.4. Statistical Validation of the Mathematical Models Developed to Predict the Extracted Amount of PP Total Polyphenols and PP Total Flavonoids by Aqueous VMAE

The validation of two derived models for predicting the total PP polyphenols and total PP flavonoids was carried out by paired t-test between the predicted and measured values of them and the results are presented in Table 4.

According to the data presented in Table 4, the differences between the measured and predicted values of the two respective parameters PP TPE and TFE are not significant and this is another proof towards the soundness of the two derived models.

Finally, in order to obtain the ultimate proof for the prediction effectiveness of the derived models we carried out two measurements at the optimum values of the operating parameters for extraction of total PP polyphenols as well as total PP flavonoids and the obtained values of TPE and TFE (each one average triplicate determination) were compared with the calculated values by the two derived RSM models. This comparison showed that (a) in the case of PP total polyphenols a 4.31% difference was found between the experimental and predicted value while (b) as far as total flavonoids the experimentally obtained value at the optimum conditions was 5.22% higher than the predicted. From the magnitude of two above mentioned differences, it is concluded, once more, that the optimization of VMAE of PP was successful.

## 3. Discussion

### 3.1. The Effect of Process Parameters on the Extracted Amounts of PP Total Polyphenols and Flavonoids by Industrial Scale VMAE Extraction

In Appendix A, the effect of the individual extraction parameters A = microwave power (W), B = water to raw PP ratio and C = extraction time (min) on the amount of PP TPE (mg GAE/2 kg of raw PP) is illustrated. From the graphs it is concluded that: (a) as the parameter A = microwave power (W) increases the amount of PP TPE is increased accordingly up to a maximum value is reached and then after that declines; (b) concerning the parameter B = water to raw PP ratio, as this increases it is observed an increase of the amount of PP total TPE up to the maximum limit of the design space (B = 30); (c) The effect of the parameter C = extraction time on the amount of PP TPE can be described as follows: initially we have a sharp increase or the TPE vs. time which, after a certain time has elapsed, this increase is getting less sharp and finally a maximum is achieved very close to the upper limit of the design space (approx.. at extraction time 119 min). Concerning the model terms which have statistically significant effect on the value of the amount of PP TPE, according to ANOVA applied to the derived model, they are the following: A, B, AB, AC, BC, A^2^, B^2^, C^2^, ABC, A^2^, AB^2^, AC^2^, C^3^.

In Appendix A, the effect of the individual extraction parameters A = microwave power (W), B = water to raw PP ratio and C = extraction time (min) on the amount of PP total flavonoids (TFE) (mg QE 2 kg^−1^ of raw PP) is illustrated. From the graphs it is concluded that: (a) as the parameter A = microwave power (W) increases the amount of PP TFE is also increased until a maximum value is reached and then after that point it declines (b) concerning the effect of the parameters B = water to raw PP ratio and C = extraction time on the amount of PP TFE it is similar to the above described for the parameter A. Concerning model terms which have statistically significant effect on the value of the amount of PP TFE, according to the ANOVA which performed to the derived model, they are the following: A, B, AB, AC, BC, A^2^, B^2^, C^2^, ABC, A^2^B, C^3^.

Furthermore, regarding the conditions required to achieve maximum amount of PP TPE and TFE, the results of the optimization for each one of the two responses respectively show that the maximum amount of PP total polyphenols (maximum TPE) is obtained at higher microwave power and also higher water to raw PP ratio as well as extraction time than TPE. However, there is a set of extraction conditions that can be a reasonable compromise between the conditions that individually optimize TPE and TFE and at which both the abovementioned response are simultaneously optimized. At these conditions and in particular for: A = Microwave power (W) = 3807.85 W, B = Water to raw PP ratio = 21.68 and C = t = Extraction Time (min) = 64.5 min, the optimal values of TPE and TFE obtained by simultaneous optimization of them are very close to the respective optima of TPE and TFE (for TPE 91.2% and for TFE 95.3% of the individual optimum respectively).

### 3.2. The Effect of Process Parameters on the Productivities of the Industrial Scale VMAE Extraction of Raw PP Total Polyphenols

With regards to the effect of A = Microwave power (W) on the rate of VMA extraction of PP total polyphenols and according to the trend illustrated in Appendix A, initially as the A parameter increases, there is a very smooth reduction of the rate of PP total polyphenols extraction which in a later stage becomes very sharp. In the second graph of Appendix A as the B parameters increases the extraction rate of PP total polyphenols increases until a maximum values is reached and then, after that, it declines. Finally, as the C = Extraction time factor is increased, a sharp reduction of the value of the rate of PP total polyphenols extraction occurs. These observations imply that in order to have high extraction rate and thereafter high productivity of PP total polyphenols at industrial scale an operation at middle microwave power and water to raw pomace ratio and for a very short time (only 15 min) has to be employed. On the other hand, according to the ANOVA applied to the derived model which correlates the rate of PP total polyphenols VMAE with the A,B,C extraction factors, the statistically significant model components and interactions are the following: A, B, C, AB, AC, BC, A^2^, B^2^, C^2^, ABC, A^2^B, AB^2^, AC^2^, C^3^.

Moreover, in Appendix A, the effect of the individual extraction parameters A = microwave power (W), B = water to raw PP ratio and C = extraction time (min) on the rate of extraction of PP flavonoids is illustrated. From the graphs in Appendix A it is concluded that: (a) As the A factor increases, the rate of extraction of PP flavonoids increases until a maximum value is approached and after that it declines; (b) the effect of B = water to raw PP pomace ratio factor on the extraction rate of PP flavonoids is following the same trend as with A as an initial increase towards a maximum value followed by a decrease after this maximum is observed; (c) the effect of the extraction time (C parameter) on the VMA extraction rate of PP flavonoids is that as the extraction time is increased the extraction rate of PP total flavonoids is reduced. Furthermore, according to the ANOVA performed to the derived model, which correlates the VMA extraction rate of PP total flavonoids to the extraction factors A,B,C, the statistically significant model components and interactions are the following: A, B, C, AB, AC, BC, A^2^, B^2^, C^2^, ABC, A^2^B.

In addition, concerning the optimum values of the extraction parameters which maximize the rates of extraction of PP total polyphenols and total flavonoids there is a considerable difference between them as the optimum value of the rate of PP total polyphenols is obtained at 2048.62 W while the corresponding maximum for the rate of PP total flavonoids extraction is obtained at microwave power 4008.62 W. In addition, the extraction ratios corresponding to the maximum values of the extraction rates of PP total polyphenols and total flavonoids respectively, are quite different. In specific, in the case of PP total polyphenols the value of water to raw PP ratio is B = 23.11 and thus higher than the one in the case of the total flavonoids (B = 18.08). On contrary, the extraction times for optimum rates of PP total polyphenols and total flavonoids extraction are small and almost identical and approx. 15 min.

### 3.3. Comparison of the Optimized Values of Extracted PP Total Polyphenols and Flavonoids of the Present Research Work with the Corresponding Results of Previous Works

During the last decade, there have been numerous literature references dealing with predictive modeling and optimization concerning the extraction of natural antioxidants from pomegranate peel that is produced in vast quantities by the pomegranate juice industry. However, literally all of them have conducted using lab scale equipment and most of them were carried out by employing non-green solvents and dried pomegranate peel powder as raw material. Furthermore, there is not any application to utilize the whole pomegranate juice industry solid waste in raw form, which is known as pomegranate pomace (PP), and in addition there is not, according our knowledge, any research work on the application of the VMAE towards the production of high added value natural antioxidants from PP at an industrial scale.

Magangana et al. [39], in their recent and comprehensive review on the various factors affecting the phytochemical and nutritional properties of the pomegranate waste, have outlined the various research efforts that have been paid worldwide in order to optimize the yield of the extraction of the natural antioxidants from pomegranate peel.

Fawole et al. [40] studied the methanolic extraction of pomegranate peel by 80% (*v/v*) methanol and distilled water. High amounts of phenolic compounds were found in peel extracts, with the highest total phenolic content (TPC) of 295,500 mg kg^−1^ dry PPL found in Ganesh and the lowest in Molla de Elche cultivar, 179,300 mg kg^−1^ dry PPL.

In addition, Pan et al. [41], experimented with aqueous ultrasound-assisted extractions in continuous (CUAE) and pulsed modes (PUAE) and compared the results obtained by these novel technologies with the ones of convectional extraction (CE) using as raw material dry pomegranate peel from fruits of Wonderful variety. The main conclusion was that pulsed ultrasound-assisted extraction (PUAE) increased the antioxidant total polyphenols yield to 148,000 mg GAE kg^−1^ dry PPL and manage to decrease the extraction time by 87% in comparison to convectional extraction (CE).

Castro-López et al. [42] determined the influence of the extraction method and solid–liquid ratio on the total phenolic content, as well as the antioxidant abilities of four plant materials, namely *Punica granatum* peels, *Juglans regia* shells, *Moringa oleifera*, and *Cassia fistula* leaves. Out of the samples tested, the pomegranate peel extracts using microwave-assisted extraction method had the highest total phenolic content value, which was measured to be 18,920 mg GAE kg^−1^. Zheng et al. [43] carried out aqueous microwave assisted extraction of dry pomegranate peel and the average experimental phenolic yield under the optimum conditions was found to be 210,360 ± 2850 mg GAE kg^−1^ of dry PPL.

Wang et al. [5] investigated the solvent extraction of pomegranate peel from Wonderful fruit variety by a series of solvents (water, methanol, acetone, ethanol, ethyl acetate) at 40 °C, 15:1 solvent to solid ratio and 4 h extraction time and they concluded a maximum extraction yield 46,510 mg GAE kg^−1^ dry PPL. Shiban et al. [44] studied the extraction of dry PPL by 80% methanol (MeOH) and found out that the highest yields for total phenolic content (TPC) and total flavonoid content (TFC) were 274,000 mg gallic acid equivalent (GAE) kg^−1^ and 56,400 mg rutin equivalent (RE) kg^−1^ of PPL (equal to 35,250 mg QE kg^−1^PPL) respectively. Nag et al. [45] cited that, dry PP and combined organic solvent extracts presented total polyphenols and total flavonoids contents 249,400 mg GAE kg^−1^, 59,100 mg rutin kg^−1^ of dry PPL (or 36,938 mg in QE kg^−1^ dry PPL) respectively. Li et al. [9] applied supercritical CO_2_ extraction of the dry pomegranate peel with the highest yields, 10,010 mg GAE kg^−1^ dry PPL to be obtained at 200 and 300 bar, 40–50 °C, and addition of 20% co-solvent.

Yasoubi et al. [46] and Mushtaq et al. [47] investigated the enzyme-assisted SCF extraction of PPL with ethanol as co-solvent and achieved total polyphenols 310,530 mg GAE kg^−1^ dry PPL In addition, Kazemi et al. [48] studied the enzymatic ultrasound-assisted extraction of polyphenols and flavonoids from dry PPL and observed an optimum extraction yield of 19,770 mg gallic acid equivalent (GAE) kg^−1^, 17,970 mg quercetin equivalent QE kg^−1^ for total polyphenols and total flavonoids respectively. This was achieved at an ultrasonication time of 41.45 min, enzyme concentration of 1.32 mL/100 mL, incubation time of 1.821 h, and incubation temperature of 44.85 °C.

Moorthy et al. [49] employed ultrasound-assisted extraction with ethanol as solvent to extract phenolics from dry pomegranate peel. In the context of this research work it was observed that the optimal extraction process conditions were as follows: extraction time of 25 min, ethanol concentration of 59%, solid-to-solvent ratio of 1:44, and extraction temperature of 80 °C. In addition, the total phenolic content (TPC) values in the obtained extracts by using ultrasound-assisted extraction (UAE) technique varied between 81,610 and 190,940 mg GAE) kg^−1^ dry weight (DW). Alexandre et al. [50] investigated the pressure assisted extraction of dry PP. Finally, Wijngaard et al. [51], experimented on ultrasound-assisted pressurized liquid extraction (UAPLE) with a solvent systems of plain ethanol + water 30, 50, and 70% *v/v* and concluded that by using a larger dry peel particle size of 1.05 mm, water extraction, extraction temperature of 70 °C, ultrasound power of 480 W, and three cycles, an enhanced phenolic recovery yield of 61,720 ± 7700 mg kg^−1^ was achieved from the pomegranate peel.

From the aforementioned literature information it is shown that the optimum yield of the total polyphenols extraction from dry pomegranate peel varies substantially among the several extraction methods and it can be found in the range from 10,010 mg GAE kg^−1^ dry PPL to 310,530 mg GAE kg^−1^ dry PPL In addition, in the case of the microwave assisted extraction this value is about 210,000 mg GAE kg^−1^ dry PPL which is similar to the obtained optimum value of 209,703 mg GAE kg^−1^ of dry PP in the context of the present research work but with a very important observation that our results are concerning industrial and not lab scale application. In addition, it is worth noting that, according to the data of the literature review, it seems that the application of VMAE with raw pomegranate pomace at industrial scale has applied for the first time in the context of the present work, offering this way the advantage of direct commercial applicability. Furthermore, the performance of the aqueous VMAE of PP observed in our work is higher for the most of the previous works even though in our case water was used as solvent instead of the more effective but non-green organic solvents.

Concerning the obtained optimum PP flavonoids extraction yield the value obtained in the present work, 21,938.3 mg QE kg^−1^ of dry PP, appears to be lower than the corresponding PP flavonoids reported by Shiban et al. [44] and Nag et al. [45], 35,250 mg QE kg^−1^ PPL and 36,938 mg in QE kg^−1^ dry PPL respectively but we have to consider that in their case methanol and other organic solvents instead of green water solvent were used, which make the results not directly comparable.

In conclusion, the magnitude of the optimum values of extracted total polyphenols and flavonoids by VMAE PP in this work prove that aqueous VMAE represents a viable alternative to the conventional extraction methods used for industrial scale extraction of polyphenols and flavonoids from PP, providing the advantages of a green process with low energy consumption and reasonable cost of initial investment.

### 3.4. Summary of the Points of Novelty of the Present Research Work

Concerning the novelty of the present research this is based on the following points: (a) Development of a completely new two-step modeling approach of modified RSM optimization with substantially improved accuracy and of general use; (b) Use of a multi-point “historical data” experimental design formed by 73 experimental points in total in order to improve the accuracy of the derived predictive models and the precision of the determination of the optimum values; (c) Predictive modeling and Optimization of PP extraction by aqueous “green” vacuum microwave-assisted extraction (VMAE) and for the first time at industrial scale and variable temperature mode; (d) Dual optimization target (i) maximum recovery of PP antioxidants and (ii) maximum economic performance; (e) Use as raw material of raw pomegranate pomace against dried PP which has been used in previous research works and therefore avoidance of the costly and quality deteriorating PP drying and milling pre-treatments.

### 3.5. Comparison of the Models Derived in the Present Work with Models Suggested in the Literature

Vacuum-microwave-assisted extraction is a typical paradigm of a solid–liquid extraction and many researchers have developed models for the prediction of the concentration or alternatively of the amount of the extracted bioactive compounds. In general, there are two types of mathematical models dedicated to microwave extraction: theoretical models based on chemical engineering principles of solid–liquid diffusion [52,53,54,55] and empirical statistical models based mostly on response surface methodology (RSM) [56,57,58,59,60] but also on adaptive neuro-fuzzy inference system (ANFIS) statistical methodology [61,62]. From the abovementioned models, the theoretical ones are not very useful in the case we need to obtain the optimum conditions for industrial-scale optimization of the VMAE extraction of bioactive phytochemicals. This is because they involve only the dependence of the extracted amount of the targeted substance vs. time and not the combined effect of the three significant extraction parameters (microwave power, water to solid ratio and extraction time). On the contrary, the empirical statistical models correlate the extraction yield to all extraction parameters and they can provide the overall optimum of the response. Furthermore, the empirical models incorporate the effect of the disintegration of a part of the total polyphenols and flavonoids due to shear or thermal stress during the process which is not taken into account in all of the abovementioned theoretical models. For the above reason, in our case, a novel empirical model was proposed based on the RSM methodology but with significant differences compared to the typical Box and Behnken experimental design used by previous researchers. In particular, with the target to increase the accuracy of the optimization a novel modeling approach was adopted according to the following points: (a) a denser experimental plan designated as “historical data design” was selected with a significantly larger number of experimental points compared to the well-known Box & Behnken design used by other researchers; (b) A novel two step optimization was adopted and in particular initially modified responses were formulated by dividing the amounts of TPE and TFE by t and t^2^ to obtain optimum simulation with high R^2^, by reducing non linearity, and then the classical RSM optimization was applied in predicted values on the basis of the improved models derived in this manner and (c) by using the Cox & Box plot provided by the statistical software (Design Expert), a proper transformation of the model response was selected in order to improve the model accuracy and a more accurate cubic instead of quadratic polynomial model was used to effectively fit the experimental data. This novel methodology results an improvement of our previous modelling and optimization attempt concerning the optimization of industrial VMAE of orange pomace [60]. The increase of the degree of the polynomial model from 2 to 3 is suggested by the Design Expert software in cases where there is a strong nonlinearity in the dependence of the response (amount of extracted phytochemical) on the extraction parameters. This high nonlinearity can be easily concluded if we observe the form of theoretical model equations given in the literature above. However, the novel approach of predictive modeling and optimization given in this paper can be tested further with more applications in order to be validated. Then it can be used to increase the effectiveness of the derived empirical statistical models dedicated to microwave extraction, by using a larger number of experimental points which can eliminate the negative effect of a single erroneous measurement and by proper transformation of the response. Finally, despite the fact the experimental effort is heavier by this approach because of the higher number of experiments and the industrial size, the much better precision and the avoidance of scale-up can pay back for the extra effort.

## 4. Materials and Methods

### 4.1. Pomegranate Pomace

The pomegranate pomace was kindly supplied by the Greek pomegranate juice producer, Alberta S.A. which is established in Argos Peloponnese-Greece. The pomegranate variety from which the obtained pomace was coming from was the well-known pomegranate fruit variety “Wonderful” and its moisture content was 67% *w/w*. The obtained pomace was passed through a commercial meat mincer (model Candy Comet supplied by D. Tomporis Co., Larisa, Greece) with a 3 mm hole diameter screen in order to become comminuted in rod shape of 3 mm diameter and then it was kept in properly sealed vacuum plastic bag (2 kg per each bag) at −25 °C until used for extraction. Drying was not applied to the pomegranate pomace in order to avoid oxidative degradation of the bioactive compounds.

### 4.2. Description of the Microwave Extractor and of the Extraction Methodology

The extraction of the pomegranate pomace samples were conducted by using the industrial scale vacuum microwave extractor model MAC−75 (Milestone Inc., Sorisole (BG)–Italy) which is established in the premises of Pellas Nature Co (Edessa, Greece) and illustrated in Figure 5.

The extraction trials of the pomegranate pomace samples were conducted following the procedure described below. The frozen pomegranate samples were first thawed at ambient temperature and 2 kg of each sample were then collected and used as the extraction sample. The 2 kg pomegranate pomace sample was first put in a plastic basket which then adjusted to the in the extraction cavity of MAC−75 Vacuum Microwave Extractor. Consequently, the machine door was closed and filled with the appropriate quantity of distilled water. The quantity of the water used in each trial was according to the water/solid ratio suggested by the experimental plan (shown below). In addition, the desired values of microwave power and extraction time were set via the electronic panel of the extractor, according to the experimental plan, and the industrial scale extractor was set in automatic operation. Cooling was not used during the extraction period and the temperature set point was set to maximum 80 °C. (adoption of a rising temperature extraction mode of operation). During each extraction trial the samples of the extracts were collected at regular time intervals (in particular 15, 30, 45, 60, 75, 90 and 120 min), filtered through plain filter paper and the filtrates were collected in plastic bottles and coded accordingly in order to easily and safely distinguish different samples. The collected samples were kept frozen at −25 °C in the freezing facility of the Laboratory of Food and Biosystems Engineering (University of Thessaly, Larisa, Greece) for a short period until the selected bioactivity parameters were analyzed.

### 4.3. Total Polyphenols Determination Method

For the determination of the total polyphenols as GAE (gallic acid equivalents) of the obtained pomegranate extracts, a slightly modified version of the method of Singleton et al. [63] and Waterhouse [64] was used. According to this method, initially a gallic acid solution was prepared by dissolving 0.5 g gallic acid in 10 mL pure ethanol and the solution was then transferred in a 100 mL volumetric flask and the rest of the volume was filled by distilled water (preparation of a gallic acid stock solution of 5000 ppm). In addition, in a 1 L glass beaker, 200 g of anhydrous sodium carbonate were dissolved in 800 mL distilled water and the solution was boiled until the salt was fully dissolved.

The solution was then cooled and kept at 24 h in dark, which resulted in the formation of crystals of anhydrous sodium carbonate, which were removed by filtration the next day. The clear filtrate was finally dissolved in a total volume of 1 L by adding the remaining distilled water in a 1L volumetric flask. Consequently, a set of standards of gallic acid was prepared by diluting 0, 1, 2, 3, 5, 10 and 20 mL of the gallic acid stock solution in six volumetric flasks of 100 mL each and filled with distilled water up to 100mL volume in order to prepare standard solutions of 0, 50, 100, 150, 250, 500 and 1000 ppm gallic acid. From each standard solution a quantity of 20 μL was mixed with 1.58 μL distilled water and 100 μL Folin Ciocalteu reagent in a glass tube and within 8 min a quantity of 100 μL sodium carbonate solution was added and the tubes were incubated for 2 h at 20 °C, after which their absorbance was measured by a UV-Vis photometer (model EVOLUTION TM 201 supplied by Thermo-Scientific Co, Shanghai, China) against the blind solution (0 ppm gallic acid concentration). The standard curve depicting gallic acid concentration vs. absorbance was constructed using the Microsoft Excel software and its R^2^ value was 0.9982. Calculation of the total polyphenols of extracts of pomegranate pomace was carried out following the same procedure and using the following equation of the standard curve:Total polyphenol concentration of extract in ppm of GAE = Absorbance of sample at 765 nm/0.001(9)

Each measurement concerning total polyphenols was carried out in triplicate and the result was the average of the three obtained values.

### 4.4. Total Flavonoids Determination Method

The total flavonoids content expressed as mg of quercetin equivalents (QE)/L of the obtained pomegranate pomace extracts was determined by using the colorimetric method of AlCl_3_, as described by Chandra et al. [65]. The method is based on the principle that AlCl_3_ reacts with the hydroxyls of the flavonoids and produces a colored complex which has maximum absorbance at 420 nm. The total flavonoids content was expressed as quercetin equivalents (QE) per L of extract. The determination method for the total flavonoids was carried out as below: 1.0 mL of the pomegranate pomace extract or standard solution (used for the construction of the calibration curve) was added in a glass test tube to which 3 mL methanol, 200 μL of aqueous solution of 10% *w/v* AlCl_3_, 200 μL 1Μ potassium acetate solution and 5.6 mL distilled water were added. The tube was then agitated by vortex and incubated for 30 min at ambient temperature for the completion of the chemical reaction. The absorbance of each sample was measured at 420 nm against a blind solution which contained all the reagents except for the pomegranate pomace extract which was replaced by distilled water.

For the construction of the calibration curve, a quercetin stock solution of 1000 ppm was prepared as well as a series of standard solutions of 50, 100, 200, 500 and 1000 ppm by serial dilutions of the stock. The absorbance of standard solutions was measured and plotted against their concentration and the linear equation obtained by Excel was used for the determination of the concentration of the total flavonoids of the pomegranate pomace extracts. The R^2^ value of the obtained linear correlation was 0.9834.

Calculation of the total flavonoids of extracts of pomegranate pomace was carried out following the same procedure and using the following equation of the standard curve:Total flavonoids concentration mg QE/L of extract = Absorbance ofsample at 420 nm/0.0055(10)

Each measurement concerning total flavonoids was carried out in triplicate and the result was the average of the three obtained values.

### 4.5. Chemicals Used for Antioxidant Tests

All the chemicals used for the abovementioned antioxidant tests were selected from the standard catalog of the Sigma Aldrich company and supplied by the Greek representative Life Sciences Chemilab (Thessaloniki, Greece).

### 4.6. Modeling and Optimization Methodology

The statistical software Design Expert 7.0.0. by Stat-Ease Inc., Minneapolis, USA was used for RSM modeling and optimization in all cases in the context of the present research work.

The methodology used for modeling and optimization had the following aims:Modeling and optimization of total pomegranate pomace polyphenol extraction.Modeling and optimization of total pomegranate pomace flavonoids extraction.Simultaneous optimization of total pomegranate pomace polyphenols, total flavonoids.Modeling and optimization (maximization) of the rate extraction of pomegranate pomace polyphenols to achieve the maximum productivity (economic optimum)Modeling and optimization (maximization) of the rate of extraction of pomegranate pomace flavonoids to achieve the maximum productivity (economic optimum)

A multi-point historical data experimental design employing 73 experimental points evenly spread in the design space was used along with response surface methodology (RSM) to derive the relevant mathematical models for the prediction of amounts of total polyphenols and total flavonoids of the pomegranate pomace extracts as well as the rates of their production and obtain the corresponding optimum values. Three factors were used as optimization factors and in particular: (a) the microwave power in the range of 2000 to 6000 W; (b) the ratio of extraction water to pomegranate pomace in the range from 10 to 30 and (c) the extraction time in the range of 15 min to 120 min and five optimization responses: (a) amount of the extracted total PP polyphenol; (b) amount of the extracted total PP flavonoids; (c) amount of the extracted total polyphenols and total flavonoids content simultaneously; (d) the rate of the extraction of the total pomegranate pomace polyphenols (e) the rate of extraction of the total pomegranate pomace flavonoids. The Design Expert 7.0.0 statistical software was used to preform predictive modeling and Optimization and derive the mathematical models. The selection of the appropriate order for the polynomial models to fit the experimental data was based to the statistical evaluation tools of the Design Expert Software. In addition, concerning the optimum response transformation in order to obtain satisfactory fitting this was selected in all cases according to the suggestion of the Cox & Box Diagram provided by Design Expert 7.0.0. The reliability of the obtained models was validated by statistical analysis (ANOVA) and in all cases the statistical significance of the derived models as well as the desirable non-significance of lack of fit were confirmed.

In the context of the present study, a modification was applied to the classical RSM modeling and in particular:

Instead of using directly the values of extracted total polyphenols (TPE) and flavonoids (TFE) and apply the RSM methodology, which in case of high non-linearity leads to polynomial models which are not able to fit successfully the experimental data, the simulation of the experimental data was based on the above mentioned responses divided by the extraction time (t in min) and by the squared value of the extraction time (t^2^ in min^2^) for the total polyphenols and total flavonoids respectively.

After this modification, the derived new responses TPE/t and TFE/t^2^ were found to be fitted very successfully by the polynomial formula of the RSM models as the non- linearity was substantially reduced.

Consequently, by using the derived RSM models, the calculated values of the TPE/t and TFE/t^2^ responses were used in order to re-calculate the values of the original responses TPE and TFE respectively and the RSM methodology was applied again to obtain the optimal values of the extraction factors as well as the corresponding values of maxima for TPE and TFE respectively.

Furthermore, the corrected values of the TPE and TFE were used for the calculation of the extraction rates of the PP total polyphenols and total flavonoids and consequently these two responses were optimized (maximized) by RSM to obtain the optimum conditions for maximum economic performance.

By the abovedescribed novel, modified RSM methodology the contribution of high order components to the responses was introduced and thus improved the simulation of the experimental data by RSM methodology which in its classical form uses only primer components to simulate the data. It is also important to mention, that in the course of the procedure described above a cubic RSM polynomial models were used instead of the second order ones typically used. This is because it is very important for successful and high accuracy optimization to use models with as high as possible correlation coefficient R^2^.

### 4.7. Determination Method of the Extraction Rates of PP Total Polyphenols and Flavonoids

The most interesting target of optimization for the industry, on economic grounds, is the maximization of the rate of extraction of polyphenols or flavonoids from raw pomegranate pomace.

The rate of the extraction of either total pomegranate pomace polyphenols or total pomegranate pomace flavonoids is determined using the following equations:Extraction Rate of OP Total Polyphenols (RTPE) (mg GAE kg^−1^ OP min^−1^) = (Amount of extracted OP total polyphenols in mg GAE)/(OP mass in kg) × (t + t_delay_ in min)(11)
where in our case the amount of the extracted total pomegranate pomace polyphenols as well as the corresponding extraction time (t) are given in Table 1. The mass of the extracted raw PP was in all cases equal to 2 kg. Furthermore the delay time (t delay) between successive extraction cycles was 15 min.

In a similar manner:Extraction Rate of OP Total Flavonoids (RTFE) (mg QE kg^−1^ OP min^−1^) = (Amount of extracted OP total flavonoids in mg GAE)/(OP mass in kg) × (t+ t _delay_ in min)(12)
where the amount of the extracted PP total flavonoids is given in Table 1 as well as the corresponding extraction times (t) and the mass of extracted raw PP was in all cases equal to 2 kg. As above, the delay time (t delay) between successive extraction cycles was 15 min.

By using the above described equations, the extraction rates of PP total polyphenols and total flavonoids where calculated and with purpose to be maximized by RSM methodology to obtain the specific conditions for economically optimum industrial scale operation.

## 5. Conclusions

In the present work, for the first time, the aqueous eco-green vacuum microwave- assisted extraction (VMAE) of phenolics and flavonoids from bioactive raw pomegranate pomace was investigated and optimized at a real industrial scale. The essential difference of the present study in comparison with previous studies is that raw pomegranate pomace was used as extraction material instead of the dried pomegranate pomace or dry peel in powder form which used in the previous studies. This can improve the economics of potential industrial production of pomegranate extracts because it omits the costly drying and milling steps involved in the case that dry pomegranate pomace or peel is used as raw material for the extraction. According to the results obtained in the present study, natural antioxidant extracts can be produced at industrial scale by eco green vacuum microwave assisted extraction from fresh pomegranate pomace at condition optimized by a novel response surface methodology based on a multi-point historical data experimental design. The optimum extraction yield for PP total polyphenols was found to be 209,703 mg GAE kg^−1^ of raw pomegranate pomace whereas for total flavonoids the optimum extraction yield was found to be 21,938.3 mg QE kg^−1^ of dry pomegranate pomace. The corresponding optimum values of the extraction parameters to achieve the abovementioned maximum polyphenol and flavonoids extraction yields were found to be: (a) microwave power = 4961.07 W, water/PP ratio = 29.90 and extraction time = 119.53 min for total polyphenols and (b) microwave power = 4147.76 W, water/PP ratio = 19.32 and extraction time = 63.32 min for total flavonoids.

Moreover, as the industrial reality demands high productivity in order to achieve cost effective production, in the course of the present work the productivities of the VMAE extraction of raw PP total polyphenols and raw PP total flavonoids were optimized and the optimum conditions for total PP polyphenols and total PP flavonoids productivities respectively were found to be: (a) microwave power = 2048.62 W, water/PP ratio = 23.11 and extraction time = 15.04 min for total polyphenols and (b) microwave power = 4008.62 W, water/PP ratio = 18.08 and extraction time = 15.29 min for total flavonoids. The optimization of the productivities can be used for financial reasons while the maximization of the amounts of the extracted PP total polyphenols and flavonoids can alternatively be useful in the case someone is putting the emphasis to the minimization of the content of phenolics in the post extraction waste. Something like that would be very important in the case this waste aims to undergo further processing (bi-refinery principle) by fermentation in order to avoid the undesirable growth delay of the used starter cultures caused by the remaining phenolics.

## Figures and Tables

**Figure 1 molecules-26-01033-f001:**
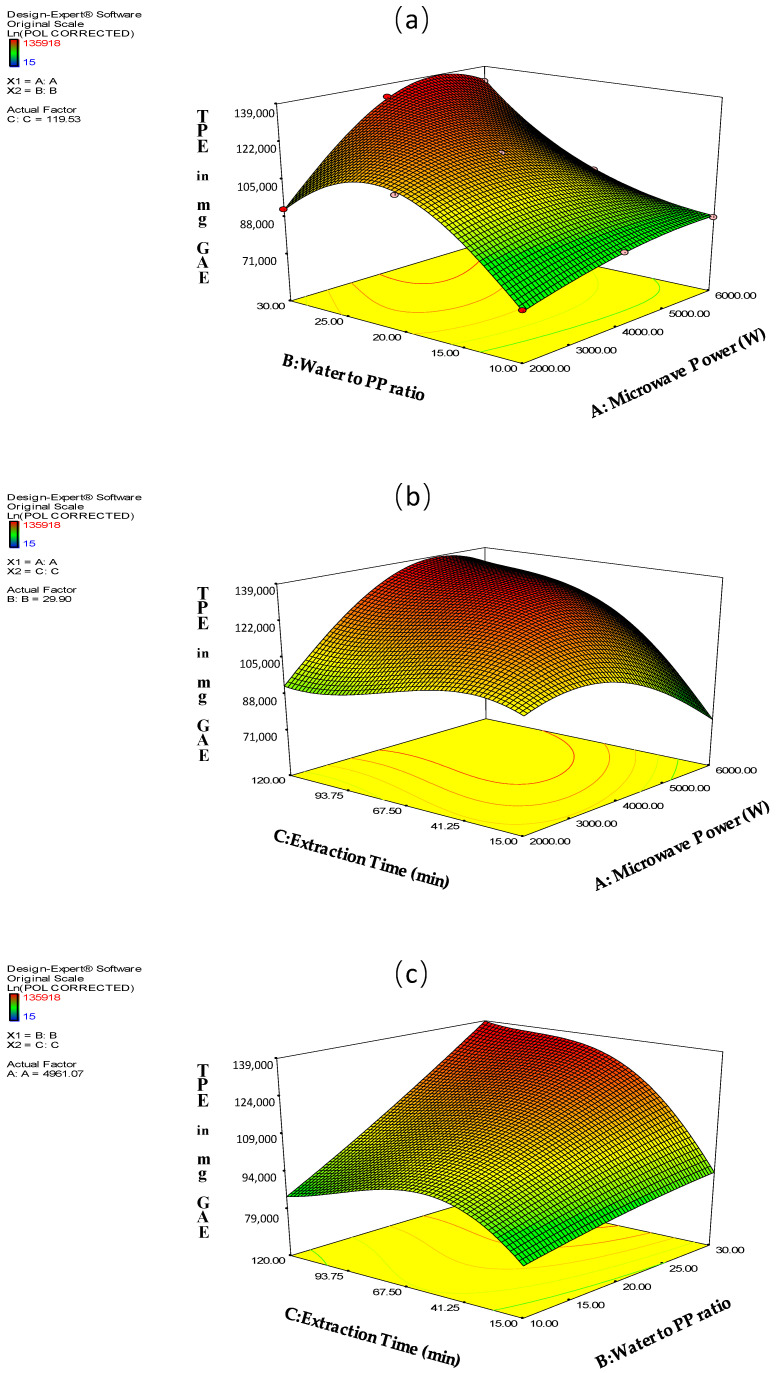
The effect of (**a**) A × B (**b**) A × C (**c**) B × C interactions on TPE response.

**Figure 2 molecules-26-01033-f002:**
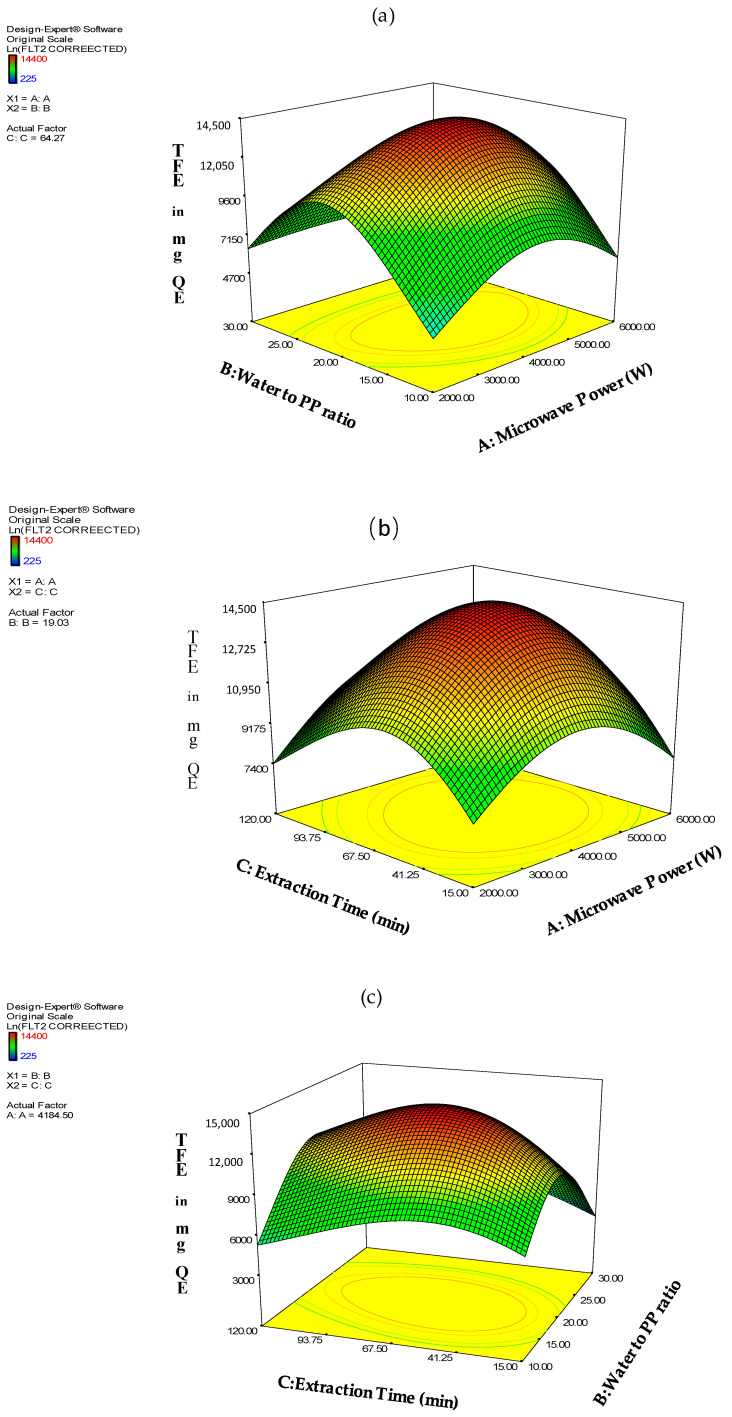
(**a**) The effects of A × B interaction (**b**) A × C interaction (**c**) B × C interaction of TFE response.

**Figure 3 molecules-26-01033-f003:**
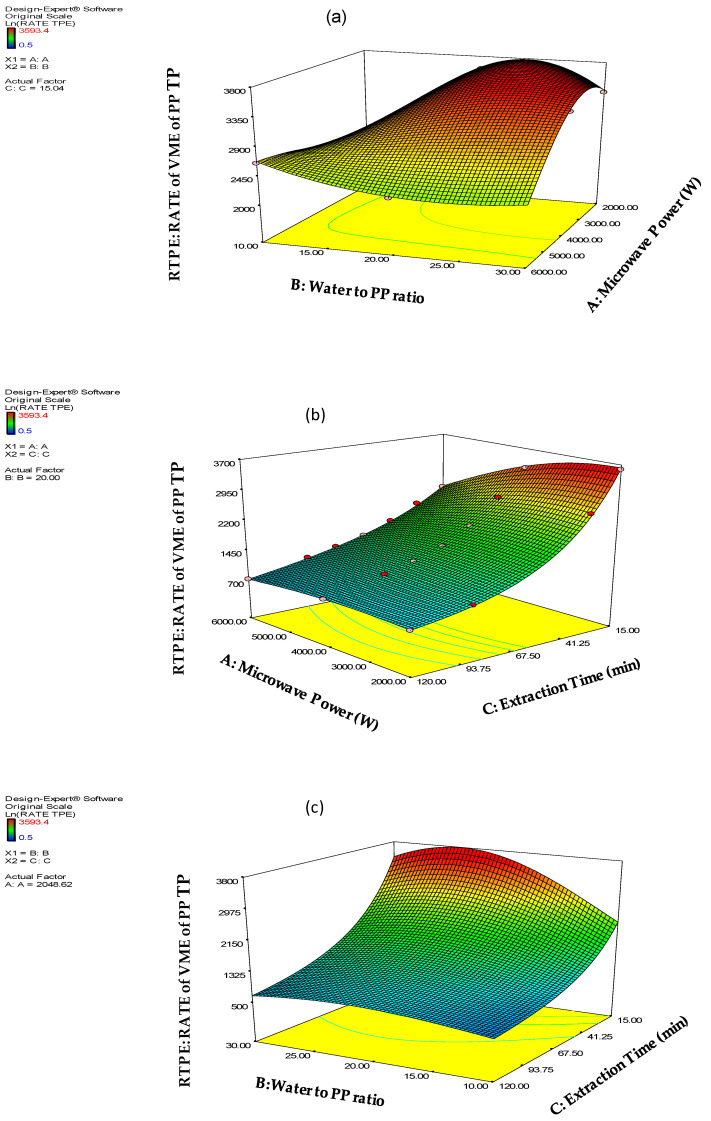
The effects of (**a**) A × B interaction (**b**) A × C interaction (**c**) B × C interaction of the rate of PP Total polyphenols VMA extraction expressed in mg GAE Kg^−1^ min^−1^.

**Figure 4 molecules-26-01033-f004:**
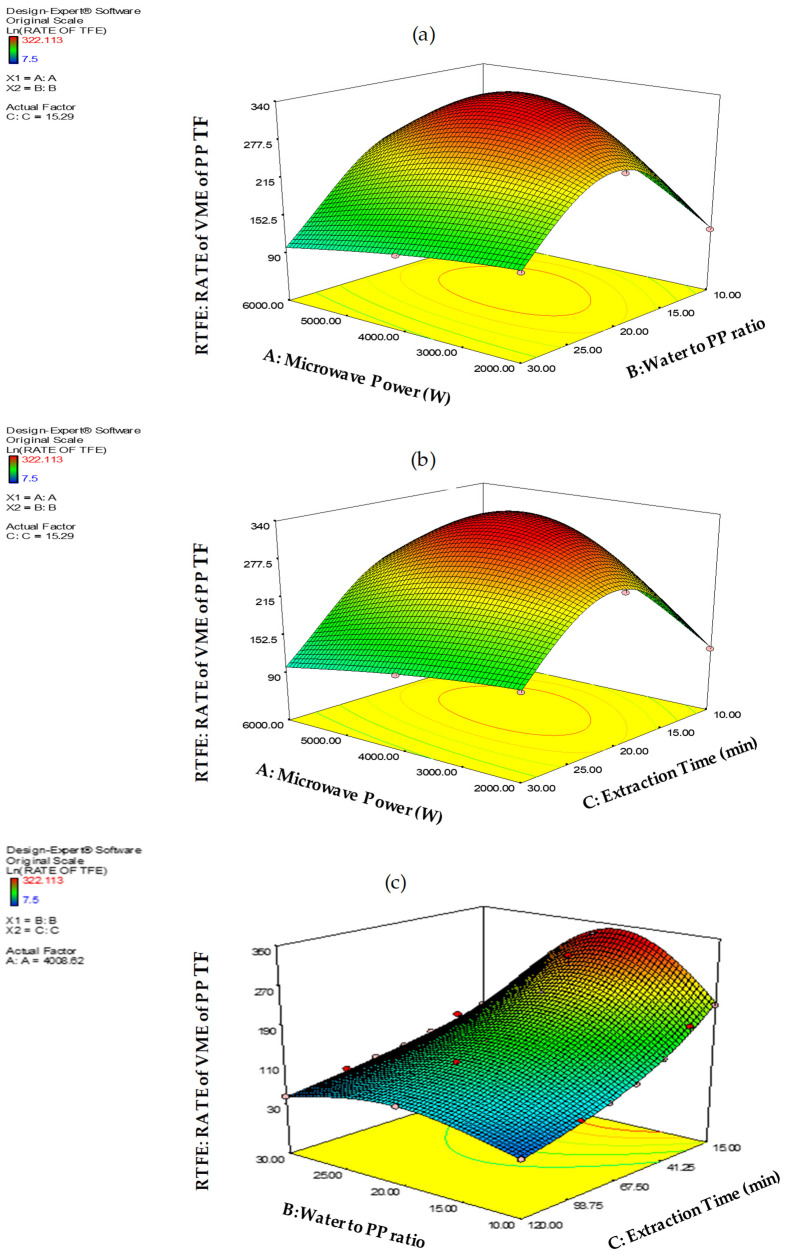
(**a**) The effects of (**a**) A × B interaction (**b**) A × C interaction (**c**) B × C interactions on the RTFE: Rate of VMAE of PP Total flavonoids expressed in mg QE kg^−1^ min^−1.^

**Figure 5 molecules-26-01033-f005:**
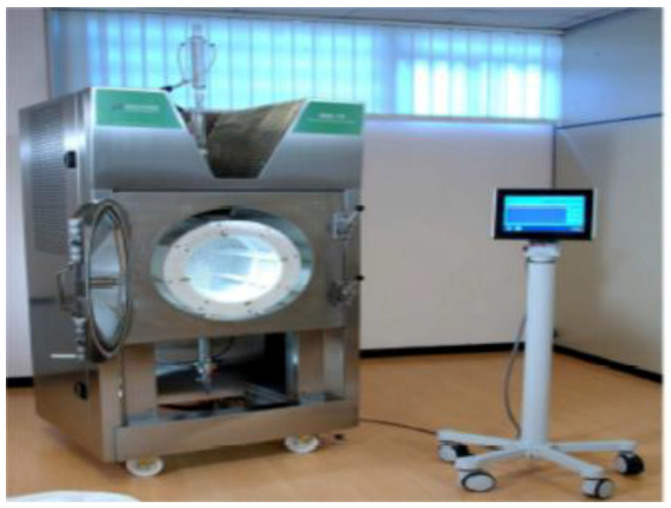
The setup of the industrial scale microwave extractor model MAC−75/Μilestone Technologies.

**Table 1 molecules-26-01033-t001:** Amount of total polyphenols and total flavonoids in raw pomegranate pomace extracts and calculated productivity indices.

A/A	Microwave Power (W)	Water to Solid Ratio (***)	Extraction Time (min)	* Amount of TPE (mg GAE) × 10^−3^	** Amount of TFE (mg QE) × 10^−3^	Cor. Rate of Extraction of PP TP (mg GAE kg^−1^ min^−1^)	Cor. Rate of Extraction of PP TF (mg QE kg^−1^ min^−1^)	Calculated Value of TPE/t (mg GAE/min)	Calculated Value of TFE/t^2^ (mg QE/min^2^)
1	4000	20	15	108.800 ± 2.300	14.1333 ± 0.432	3219.09	322.113	7253.33	62.8147
2	4000	20	30	129.600 ± 1.867	14.497 ± 0.334	2577.4	291.702	4320	16.1078
3	4000	20	45	124.000 ± 1.218	14.9333 ± 0.354	1963.17	225.649	2755.56	7.37447
4	4000	20	60	138.400 ± 1.879	14.9091 ± 0.234	1582.65	181.49	2306.67	4.14142
5	4000	20	75	128.800 ± 1.216	15.4424 ± 0.165	1336.32	155.27	1717.33	2.74532
6	4000	20	90	131.200 ± 1.934	15.8061 ± 0.765	1159.62	133.708	1457.78	1.95137
7	4000	20	120	125.600 ± 2.300	14.0606 ± 0.453	855.954	814.271	1046.67	0.976431
8	2000	30	15	106.800 ± 1.800	6.36364 ± 0.545	3317.12	149.143	7120	28.2828
9	2000	30	30	103.200 ± 1.945	6.54545 ± 0.765	2427.3	133.718	3440	7.27272
10	2000	30	45	105.600 ± 2.310	6.800 ± 0.345	1741.18	100.382	2346.67	3.35802
11	2000	30	60	104.400 ± 3.200	7.56364 ± 0.432	1321.94	79.934	1740	2.10101
12	2000	30	75	99.600 ± 1.547	5.67273 ± 0.254	1072.43	670.988	1328	1.00849
13	2000	30	90	100.800 ± 1.675	5.34545 ± 0.435	912.191	571.488	1120	0.659932
14	2000	30	120	88.800 ± 1.189	5.01818 ± 0.276	680.084	334.385	740	0.348485
15	6000	30	45	106.800 ± 2.320	5.74545 ± 0.453	2068.41	970.991	2373.33	2.83726
16	6000	30	60	126.000 ± 2.114	6.72727 ± 0.348	1741.18	830.116	2100	1.86869
17	6000	30	75	150.000 ± 3.998	7.01818 ± 0.543	1520.59	75.279	2000	1.24768
18	6000	30	90	135.600 ± 2.645	7.56364 ± 0.634	1377.02	711.768	1506.67	0.933783
19	6000	30	120	117.600 ± 1.645	6.83636 ± 0.386	1256.2	684.196	980	0.474747
20	4000	30	15	100.800 ± 2.399	2.14545 ± 0.123	974.785	514.036	6720	9.53533
21	4000	30	30	103.200 ± 3.129	3.16364 ± 0.164	3350.46	129.658	3440	3.51516
22	4000	30	45	127.200 ± 2.765	4.29091 ± 0.435	2709.54	122.209	2826.67	2.11897
23	4000	30	60	118.800 ± 2.477	6.50909 ± 0.225	2105.52	983.941	1980	1.80808
24	4000	30	75	123.600 ± 1.276	7.27273 ± 0.321	1714.47	831.961	1648	1.29293
25	4000	30	90	139.200 ± 2.865	6.32727 ± 0.342	1462.17	748.261	1546.67	0.781144
26	4000	30	120	136.800 ± 2.654	6.03636 ± 0.264	1294.46	670.648	1140	0.419192
27	2000	10	15	60.800 ± 1.288	4.41212 ± 0.114	984.582	446.882	4053.33	19.6094
28	2000	10	30	70.400 ± 1.382	5.27273 ± 0.213	2094.04	113.852	2346.67	5.85859
29	2000	10	45	74.800 ± 1.657	4.65455 ± 0.221	1578.98	101.062	1662.22	2.29854
30	2000	10	60	73.200 ± 1.764	4.53333 ± 0.432	1167.15	758.669	1220	1.25926
31	2000	10	75	71.200 ± 2.005	4.52121 ± 0.276	913.11	598.117	949.333	0.803771
32	2000	10	90	66.800 ± 1.976	4.38788 ± 0.206	763.321	496.584	742.222	0.541714
33	2000	10	120	75.600 ± 1.645	4.33939 ± 0.321	669.044	419.156	630	0.301347
34	4000	10	15	70.400 ± 2.134	4.72727 ± 0.437	529.65	242.814	4693.33	21.0101
35	4000	10	30	80.800 ± 1.287	5.38182 ± 0.239	2482.08	215.919	2693.33	5.9798
36	4000	10	45	91.200 ± 2.345	8.4000 ± 0.423	1947.95	187.867	2026.67	4.14815
37	4000	10	60	88.800 ± 3.212	7.4303 ± 0.243	1483.74	138.238	1480	2.06397
38	4000	10	75	92.000 ± 2.345	6.54545 ± 0.431	1172.46	105.763	1226.67	1.16364
39	4000	10	90	82.400 ± 1.765	5.13939 ± 0.164	980.123	869.355	915.556	0.634493
40	4000	10	120	72.000 ± 1.745	4.89697 ± 0.153	833.68	713.544	600	0.340067
41	6000	20	15	72.800 ± 1.123	7.0303 ± 0.432	603.181	392.405	4853.33	31.2458
42	6000	20	30	80.000 ± 3.234	7.46667 ± 0.234	2291.25	241.026	2666.67	8.2963
43	6000	20	45	93.600 ± 2.314	7.41818 ± 0.179	1947.95	222.679	2080	3.6633
44	6000	20	60	97.600 ± 1.156	8.55758 ± 0.297	1591.32	173.987	1626.67	2.37711
45	6000	20	75	101.600 ± 2.165	11.2727 ± 0.439	1321.94	144.2	1354.67	2.00404
46	6000	20	90	92000 ± 2576	8.24242 ± 0.275	1150.19	125.86	1022.22	1.01758
47	6000	20	120	94.400 ± 1.346	7.46667 ± 0.437	998.096	110.571	786.667	0.518519
48	6000	10	15	83.600 ± 3.423	5.90303 ± 0.355	714.953	693.876	5573.33	26.2357
49	6000	10	30	101.600 ± 4.535	6.13333 ± 0.543	2662.05	161.564	3386.67	6.81481
50	6000	10	45	98.400 ± 2.345	6.29091 ± 0.397	2174.46	136.419	2186.67	3.10662
51	6000	10	60	96.800 ± 1.786	6.49697 ± 0.487	1689.72	983.941	1613.33	1.80471
52	6000	10	75	97.600 ± 1.435	5.23636 ± 0.345	1348.65	737.884	1301.33	0.930908
53	6000	10	90	99.200 ± 1.765	4.89697 ± 0.543	1127.41	589.781	1102.22	0.604564
54	6000	10	120	96.800 ± 2.154	4.77576 ± 0.345	939.971	477.347	806.667	0.33165
55	4000	20	15	96.800 ± 2.134	12.5333 ± 0.499	621.55	250.208	6453.33	55.7036
56	4000	20	30	104.000 ± 2.165	14.303 ± 0.543	3219.09	322.113	3466.67	15.8922
57	4000	20	45	112.000 ± 1.345	15.1758 ± 0.876	2577.4	291.702	2488.89	7.49422
58	4000	20	60	127.200 ± 2.545	15.6121 ± 0.645	1963.17	225.649	2120	4.33669
59	4000	20	75	112.000 ± 2.643	17.600 ± 0.445	1582.65	181.49	1493.33	3.12889
60	4000	20	90	105.600 ± 3.567	21.0424 ± 0.654	1336.32	155.27	1173.33	2.59783
61	4000	20	120	118.400 ± 2.145	6.01212 ± 0.345	1159.62	133.708	986.667	0.417508
62	4000	20	15	96.000 ± 1.654	13.8909 ± 0.654	855.954	814.271	6400	61.7373
63	4000	20	30	120.000 ± 3.288	14.8364 ± 0.876	3219.09	322.113	4000	16.4849
64	4000	20	45	125.600 ± 1.234	14.9333 ± 0.567	2577.4	291.702	2791.11	7.37447
65	4000	20	60	125.600 ± 1.456	15.0788 ± 0.823	1963.17	225.649	2093.33	4.18856
66	4000	20	75	118.400 ± 2.143	15.5636 ± 0.455	1582.65	181.49	1578.67	2.76686
67	4000	20	90	119.200 ± 3.215	16.0242 ± 0.774	1336.32	155.27	1324.44	1.9783
68	4000	20	120	118.400 ± 2.154	13.9152 ± 0.345	1159.62	133.708	986.667	0.966333
69	2000	20	15	115.200 ± 3.276	4.77576 ± 0.222	855.954	814.271	7680	21.2256
70	2000	20	30	100.800 ± 1.222	6.95758 ± 0.342	3593.4	253.383	3360	7.73064
71	2000	20	90	94.400 ± 2.345	6.30303 ± 0.678	2655.89	224.917	1048.89	0.778152
72	4000	20	45	112.800 ± 3.123	7.63636 ± 0.543	1059.81	942.225	2506.67	3.77104
73	4000	20	60	86.400 ± 1.325	5.79394 ± 0.386	806.107	551.308	1440	1.60943

* The Figures of this column are calculated by multiplying the total polyphenols concentration of the extract by its volume = CP (mg L^−1^) × water-to-solid ratio × 2 kg. ** The figures of this column are calculated by multiplying the total flavonoids concentration of the extract by its volume = CF (mg L^−1^) × water-to-solid ratio × 2 kg. *** Water/solid ratio is expressed in L kg^−1^ and the mass of the solids were equal to 2 kg for all experiments. **** The values in columns 4,5,6,7 are the average of triplicate determination ± SD (standard deviation).

**Table 2 molecules-26-01033-t002:** Analysis of variance (ANOVA) of the derived model for prediction of the modified response TPE/t of PP extracts and the model equation. Response: TPE/t; Transform: *Natural log*; Constant: 0; Significant Model terms: B, C, AB, AC, BC, A^2^, B^2^, C^2^, ABC, A^2^B, AB^2^, AC^2^, C^3^.

Source	Sum of Squares	df	Mean Square	F Value	*p*-Value Prob > F	
Model	29.28	14	2.09	288.18	<0.0001	significant
A-A	0.013	1	0.013	1.84	0.1797	
B-B	0.54	1	0.54	74.07	<0.0001	
C-C	2.12	1	2.12	292.37	<0.0001	
AB	0.062	1	0.062	8.59	0.0048	
AC	0.10	1	0.10	14.23	0.0004	
BC	0.035	1	0.035	4.81	0.0324	
A^2^	0.23	1	0.23	31.31	<0.0001	
B^2^	0.19	1	0.19	26.24	<0.0001	
C^2^	1.32	1	1.32	181.59	<0.0001	
ABC	0.097	1	0.097	13.43	0.0005	
A^2^B	0.043	1	0.043	5.88	0.0185	
AB^2^	0.27	1	0.27	36.73	<0.0001	
AC^2^	0.053	1	0.053	7.32	0.0090	
C^3^	0.14	1	0.14	18.69	<0.0001	
Residual	0.42	58	7.258 × 10^−3^			
Lack of Fit	0.34	44	7.657 × 10^−3^	1.27	0.3203	
not significant						
	Std. Dev.	0.085		R-Squared	0.9858	
	Mean	7.57		Adj R-Squared	0.9824	
	C.V.%	1.13		Pred R-Squared	0.9740	
	PRESS	0.77		Adeq Precision	64.474	

**Table 3 molecules-26-01033-t003:** Analysis of Variance (ANOVA) of the derived model for prediction of the calculated modified response TFE/t^2^ of PP extracts and model equation. Type of polynomial Model: Cubic model Constant: 0; Response: AMOUNT OF EXTRACTED PP TOTAL FLAVONOIDS (mg QE/2 kg raw PP/t^2^; Transform: *Natural log*; Significant Model terms: C, BC, A^2^, B^2^, C^2^, A^2^B, C^3^.

Source Model	Sum of		Mean	F	*p*-Value	
	Squares	df	Square	Value	Prob > F	
	132.01	12	11.00	157.72	<0.0001	Significant
A-A	0.080	1	0.080	1.15	0.2873	
B-B	0.13	1	0.13	1.91	0.1716	
C-C	8.53	1	8.53	122.35	<0.0001	
AB	0.062	1	0.062	0.89	0.3487	
AC	0.081	1	0.081	1.16	0.2866	
BC	0.41	1	0.41	5.88	0.0184	
A^2^	1.20	1	1.20	17.22	0.0001	
B^2^	6.90	1	6.90	98.89	<0.0001	
C^2^	4.53	1	4.53	64.94	<0.0001	
ABC	0.23	1	0.23	3.23	0.0775	
A^2^B	0.35	1	0.35	5.09	0.0278	
C^3^	0.88	1	0.88	12.59	0.0008	
Residual	4.19	60	0.070			
Lack of Fit	3.771 × 10^−3^	50	7.541 × 10^−5^	0.19	1.0000	not significant
Pure Error	0.011	28	3.913 × 10^−4^			
Cor Total	0.094	86				
Fit Statistics						
	Std. Dev.	0.26		R²	0.9693	
	Mean	1.02		Adjusted R²	0.9631	
	C.V.%	25.82		Predicted R²	0.9510	
				Adeq Precision	47.084	

**Table 4 molecules-26-01033-t004:** Results of the statistical t-test analysis between predicted and actual values of the total polyphenols and total flavonoids of PP extracts.

a/a		*t* Value	*p* Value	Statistical Significance (2-Tailed) Significance Level = 0.05
1	Pair 1 PM-PP	−0.315486	0.7533	*p* value = 0.7533 > 0.05 Therefore no significant difference between predicted and measured amounts of PP total polyphenols
2	Pair 2 FM-FP	1.127934	0.26309	*p* value = 0.26309 > 0.05 Therefore no significant difference between predicted and measured amounts of PP total flavonoids.
	(1) PM Measured values of extracted total PP polyphenols (2) PP Predicted values of extracted total PP polyphenols (3) FM Measured values of extracted total PP flavonoids (4) FP Predicted values of extracted total PP flavonoids			

## Data Availability

The data presented in this study are available on request from the corresponding author.

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
