# Peer review of "Optimization of the Vacuum Microwave Assisted Extraction of the Natural Polyphenols and Flavonoids from the Raw Solid Waste of the Pomegranate Juice Producing Industry at Industrial Scale"

_molecules, 2021, doi:10.3390/molecules26041033_

Round 1

Reviewer 1 Report

General Comments 

The objective of the research paper entitled ‘Optimization of the vacuum microwave assisted

extraction of the natural polyphenols and flavonoids from the raw solid waste of the pomegranate juice producing industry at industrial scale’ was to investigate and Optimize the vacuum microwave extraction of raw pomegranate pomace, for the first time at industrial scale, and with dual maximum recovery as well as maximum economic performance criteria and thus obtain the optimum extraction conditions (microwave power, water to raw PP ratio, extraction time) corresponding to each one, respectively.

The work carried out in this manuscript is good, but the entire manuscript is very lenthy, a lot of information can simply be removed. Lot of figures are un-necessary, which need to be addressed before it is accepted for publication in Molecule journal.

The abstract needs to be re-written, it's not comprehensive (written very randomly, no results are mentioned, no conclusion is drawn) and has a few grammatical mistakes too.

Introduction section is very lenthy, lot of sentences can be removed. Make it crisp and tight. No need to provide a lot of background literature.

Results section is full of unnecessary figure that is not required

Discussion section has a lot on non-required information which can easily removed, its not adding any value to the paper

Conclusion should be a paragraph or two, its very legthy.

Table and figure numbers are not in order, for instance Table 4 is coming first in the text

Other suggestions

lines 31, optimization

line 39, extraction of PP pomace… what is PP? write in full when it comes first time in introduction as well

Line 74, reported that..

Line 115-118, can be deleted, observatory sentence..

Line 123-130, once you outlined you objectives thean no need to make recommendation, can be deleted.

Line 312, please check the representation of time

Line 446, optimization..

Line 862, Check degree centigrade sign

Line 865, Check R2 sign

Line 875, 881, check AlCl3

Author Response

REVIEWER 1

The work carried out in this manuscript is good, but the entire manuscript is very lenthy, a lot of information can simply be removed. Lot of figures are un-necessary, which need to be addressed before it is accepted for publication in Molecule journal. Many Figures were transferred to supplementary materials in the revised manuscript. Also the conclusion part was reduced and the introduction was significantly cut.

The abstract needs to be re-written, it's not comprehensive (written very randomly, no results are mentioned, no conclusion is drawn) and has a few grammatical mistakes too.

The abstract was re-written following the suggestion of the reviewer and have now results and conclusion

Introduction section is very lenthy, lot of sentences can be removed. Make it crisp and tight. No need to provide a lot of background literature.

A significant part of introduction was removed and in particular: lines 45-47 , lines 115-118 and lines 123-130 following the correct remark of both reviewers

Results section is full of unnecessary figure that is not required

Many of the figures and in particular the figures1,2,4 were put in supplementary material. Corrected.

Discussion section has a lot on non-required information which can easily removed, its not adding any value to the paper

Many details in the form of Figures were transferred to supplementary2 and this way we have less details in the Discussion section.

Conclusion should be a paragraph or two, its very legthy.

Conclusion significantly cut in two paragraphs as it is suggested by the reviewer. Corrected.

Table and figure numbers are not in order, for instance Table 4 is coming first in the text Table 4 was put by mistake in line 133 of the original manusctipt. Now it is corrected to Table 1 in the revised manuscript.

Other suggestions

lines 31, optimization

corrected as the reviewer suggested from Optimization to optimization

line 39, extraction of PP pomace… what is PP? write in full when it comes first time in introduction as well

corrected to: of  pomegranate pomace (PP)

Line 74, reported that..

Corrected from “the” to “that”

Line 115-118, can be deleted, observatory sentence..

The text: “The present study is coming complementary to this investigation in order to examine the potential of using all the solid waste of pomegranate juice industry and not only the peel as raw material to produce high added value natural extracts in a cost effective way and at industrial scale.” Was removed

Line 123-130, once you outlined you objectives thean no need to make recommendation, can be deleted.

The text : “This information can be used in order to produce, at industrial scale and in financially favorable way, high added value bioactive aqueous PP extracts and thus to exploit the solid agro-food waste produced in bulk quantities by the pomegranate juice production industry. The optimized aqueous pomegranate pomace extracts can be utilized either as natural antioxidants or./ and as natural antimicrobials (extracts with maximum polyphenols and/or flavonoids content) targeting high added value food & nutraceutical, cosmetic and agro-protection applications and simultaneously leading to the improvement of the carbon footprint of the pomegranate juice industry.” was removed

Line 312, please check the representation of time

The t symbol was removed

Line 446, opt

Opt was replaced by opt

Reviewer 2 Report

The manuscript titled “Optimization of the vacuum microwave assisted extraction of the natural polyphenols and flavonoids from the raw solid waste of the pomegranate juice producing industry at industrial scale” is related to evaluation of polyphenols extraction from pomegranate pomace on industrial level using vacuum microwave assisted extraction. Authors applied comprehensive design of experiment and improved RSM technique in order to achieve better correlations between responses and variables and higher model accuracy. Overall, manuscript is well written and interesting however, there 2 major drawback which are listed below same as several minor issues.

 Lines 119-125 – Maximum yield of bioactive compound is not necessarily the optimum solution from economical point of view due to sometimes higher energy consumption (Meziane et al 2020). The cited reference is related to the essential oil recovery but general idea is the same for the phenolics as well. I could agree that estimation of energy consumption on lab scale experiment could be unnecessary because conditions might change when process is scaled up. However, if the experiments are performed at industrial level, evaluation of energy consumption is a mandatory if we talk about real process optimization that could be implemented in factories. Thus, I would strongly suggest to add a section devoted to this topic. An idea for relatively simple calculation to evaluate energy consumption and carbon footprint, not complete energy study, but still somewhat useful could be found in work of Drinic et al. 2020.

Drinić, Z., Pljevljakušić, D., Živković, J., Bigović, D., & Šavikin, K. (2020). Microwave-assisted extraction of O. vulgare L. spp. hirtum essential oil: Comparison with conventional hydro-distillation. Food and Bioproducts Processing, 120, 158-165.

Meziane, I. A. A., Maizi, N., Abatzoglou, N., & Benyoussef, E. H. (2020). Modelling and optimization of energy consumption in essential oil extraction processes. Food and Bioproducts Processing, 119, 373-389.

The authors presented several optimal conditions, i.e. for maximal TP yield, maximal TF yield, simultaneous maximal TF and TP yield, maximal extraction rate for TP and TF. However, did not perform the extraction on suggested optimal conditions which is necessary in order to experimentally validate the process optimization. Thus, I strongly suggest to perform also these trial because that is the best possible validation of any process.

Other issues:

Lines 45-47 This was an interesting data, however, following the reference there is no evidence of such claim, please add appropriate reference or remove the sentence.

Lines 100-104 Please add reference/s for the advantages of green extraction techniques

Lines 110-113 - Is there possibly a mistake claiming that Skenderidis et al. used non-dependent temperature  extraction mode when temperature was evaluated on three levels (40, 60 and 80C) in that manuscript? If not, please explain this.

Table 1 Please convert columns 5 and 6 to grams or mg * 10-3, numbers are too big and it is hard to see the differences. Decimals in columns 2,3 and 4 are irrelevant, please remove them.

Figure 1, 2 and 4 could be presented as supplementary material as manuscript has too many figures in current form.

It is not necessary to mention software version so many times in Results section, please correct it (examples Lines 144, 223, 331 etc). It is sufficient to be done once in materials and methods section.

Lines 255-285, 341-344, 458-460 and 511-513 – Please add reference for 0.2 value

Lines 736-737 – abbreviation VMAE was already introduced, please use it here as well.

Lines 814-817 – Please add also here moisture content. Additionally add average particle size as it is relevant factor in extraction optimization process.

Typos:

Line 48 – Li et al.

Line 106 - Kaderides et al.

Line 110 and 113 - Skenderidis et al. (there are potentially more typos of this type, please check the entire manuscript)

Line 127 - or/

Lines 328, 330, 338, 340, 580 – TFE - there are potentially more mistakes of this type, please check the entire manuscript.

Line 574 – A.

Line 788 - selected

Line 869 – 0.001

Author Response

REVIEWER 2

 Lines 119-125 – Maximum yield of bioactive compound is not necessarily the optimum solution from economical point of view due to sometimes higher energy consumption (Meziane et al 2020). The cited reference is related to the essential oil recovery but general idea is the same for the phenolics as well. I could agree that estimation of energy consumption on lab scale experiment could be unnecessary because conditions might change when process is scaled up. However, if the experiments are performed at industrial level, evaluation of energy consumption is a mandatory if we talk about real process optimization that could be implemented in factories. Thus, I would strongly suggest to add a section devoted to this topic. An idea for relatively simple calculation to evaluate energy consumption and carbon footprint, not complete energy study, but still somewhat useful could be found in work of Drinic et al. 2020.

Drinić, Z., Pljevljakušić, D., Živković, J., Bigović, D., & Šavikin, K. (2020). Microwave-assisted extraction of O. vulgare L. spp. hirtum essential oil: Comparison with conventional hydro-distillation. Food and Bioproducts Processing, 120, 158-165.

Meziane, I. A. A., Maizi, N., Abatzoglou, N., & Benyoussef, E. H. (2020). Modelling and optimization of energy consumption in essential oil extraction processes. Food and Bioproducts Processing, 119, 373-389.

We fully agree with the reviewer that it is true that maximum yield of bioactive compound is not necessarily the optimum solution from economical point of view due to sometimes higher energy consumption. In our specific case this is not the point and let us give a specific example in order to support our claim:

Lets try to estimate what is the revenue by operating for example at microwave power:6000 W , with water to PP ratio =30 for 45 min. On thiss case according to the data of Table 1 the polyphenol production is: 106800 mg as GAE or 106,8 g per 45+15 min=60 min by tasking account the delay between runs. In addition taking into account that the average wholesale price of pomegranate polyphenol is about 80 Euro the revenue produced is : 0,1068 Kg X 80 euro/Kg/1hr=8,544 Euro/hr. At the same time the cost of energy would be=6 Kw X0,75 hrs X0,1 Euro/Kwh=0,45 euro

So if we take into account the reduction of the revenew due to energy consumption we have a different figure =8,544-0,45=8,500 Euro/hr This means a very slight and thus negligible difference. The same will be observed at any experimental point.

According to the above analysis it seems that In our specific case  the high added value of the finished product reduces to negligible the contribution of energy and makes the production rate the only economically important parameter.

The idea to calculate carbon footprint is great but we will deal in detail in another paper involving two biowastes pomegranate and orange pomace and in order to have a complete picture we also have to deal with what is going to be done with the “dead body” of the exteaction which is the post extracted solid residual (examine various possibilities like: production of animal food, production of compost or production of high added value products by fermentation).

The authors presented several optimal conditions, i.e. for maximal TP yield, maximal TF yield, simultaneous maximal TF and TP yield, maximal extraction rate for TP and TF. However, did not perform the extraction on suggested optimal conditions which is necessary in order to experimentally validate the process optimization. Thus, I strongly suggest to perform also these trial because that is the best possible validation of any process.

We full agree , it is a very good suggestion and we have put the numerical comparison at the end of paragraph  2.4.: We wrote “Finally, in order to have the ultimate proof of the prediction effectiveness of the derived models we carried out two measurements at the optimum values of the operating parameters for extraction of total PP polyphenols as well as total PP flavonoids and the obtained values of TPE and TFE (each one average triplicate determination) were compared with the calculated values by the two derived RSM models. This comparison showed that a) in the case of PP total polyphenols a 4,31% difference was found between the experimental and predicted value while b) as far as total flavonoids the experimentally obtained value at the optimum conditions was 5,22 % higher than the predicted. From the magnitude of two above mentioned differences, it is concluded once more that the optimization of VMAE of PP was successful.

Other issues:

Lines 45-47 This was an interesting data, however, following the reference there is no evidence of such claim, please add appropriate reference or remove the sentence. This paragraph is removed.

Lines 100-104 Please add reference/s for the advantages of green extraction techniques

Two new references added

Osorio-Tobón, J.F. Recent advances and comparisons of conventional and alternative extraction techniques of phenolic compounds. J Food Sci Technol 57, 4299–4315 (2020). https://doi.org/10.1007/s13197-020-04433-2

De Castro M.D.L, Castillo-Peinado LS  Microwave-assisted extraction of food components. In: Knoerzer K, Juliano P, Smithers G (eds) Innovative food processing technologies: extraction, separation, component modification and process intensification, 1st edn. Woodhead Publishing, Duxford, 2016, 57–110. https://doi.org/10.1016/B978-0-08-100294-0.00003-1

Lines 110-113 - Is there possibly a mistake claiming that Skenderidis et al. used non-dependent temperature  extraction mode when temperature was evaluated on three levels (40, 60 and 80C) in that manuscript? If not, please explain this.

By non-dependent we mean that they used these temperature in respective experiments. On contrary in our case we used in all experiments a maximum temperature stting of 80 C in all experiments and the offset of the temperature in our case was very large and we left the microwave power to rise the temperature with limit the 80 C. Anyway, this paragraph was removed according to suggestion of another reviewer and it is not existent in the revised manuscript.

Table 1 Please convert columns 5 and 6 to grams or mg * 10-3, numbers are too big and it is hard to see the differences. Decimals in columns 2,3 and 4 are irrelevant, please remove them. This was corrected.The figures in columns 5 and 6 were converted into mg * 10-3 and the decimal points in columns 2,3,4 were removed.

Figure 1, 2 and 4 could be presented as supplementary material as manuscript has too many figures in current form.

The Figures 1,2,4 are now given as Supplementary1 following the correct suggestion of the reviewer

It is not necessary to mention software version so many times in Results section, please correct it (examples Lines 144, 223, 331 etc). It is sufficient to be done once in materials and methods section.

This was corrected and now the version of the software is reported once in the paragraph 4.6. of the revised manuscript.

Lines 255-285, 341-344, 458-460 and 511-513 – Please add reference for 0.2 value

The reference was put

Two reference are in the following papers

1) https://www.statease.com/docs/v11/navigation/anova-adj-r-squared/

2) N. Bala1, M. Napiah1, I. Kamaruddin1 and N. Danlami11 Department of Civil & Environmental Engineering, Universiti Teknologi PETRONAS, 32610 Bandar Seri Iskandar, Perak, Malaysia Optimization of Nanocomposite Modified Asphalt Mixtures Fatigue Life using Response Surface Methodology, IOP Conf. Series: Earth and Environmental Science 140 (2018) 012064 doi :10.1088/1755-1315/140/1/012064

The first one is in the Program tutotial and the second is a Proceedings paper and it is the one put in the revised manuscript.

Lines 736-737 – abbreviation VMAE was already introduced, please use it here as well. Corrected The  full name was replaced by simple VMAE

Lines 814-817 – Please add also here moisture content. Additionally add average particle size as it is relevant factor in extraction optimization process.

“in rod shape of 3 mm diameter” this phrase was put for the size and “and its moisture content was 67% w/w” for the moisture content in the patagraph 4.1 in the revised manuscript.

Typos:

Line 48 – Li et al. Corrected: a full stop was put

Line 106 - Kaderides et al. Corrected

Line 110 and 113 - Skenderidis et al. (there are potentially more typos of this type, please check the entire manuscript) Corrected

Line 127 - or/ The whole paragraph was removed and so there was not need of correction

Lines 328, 330, 338, 340, 580 – TFE - there are potentially more mistakes of this type, please check the entire manuscript. In all positions the wrong TPE was replaced by the correct TFE

Line 574 – A. Corrected

Line 788 – selected: the wrong was substituted by the correct selected

Line 869 – 0.001: comma was replaced by full stop

Round 2

Reviewer 1 Report

The authors made significant modifications in the manuscript based upon the reviewer's comments, the manuscript has improved and can be accepted for publication in the molecule journal.

Minor edits spotted:

Line 16, ‘reach’ should be ‘rich’, please check.

Line 128, check min-1, use superscript function here

Reviewer 2 Report

The authors gave good arguments on inquiries and improved the paper significantly. Decimals in Table 1 should be corrected to 2 points after a comma. Once it is corrected I would recommend the acceptance for publication in Molecules.